



# Enhancing MAX-DOAS atmospheric state retrievals by multispectral polarimetry - studies using synthetic data

Jan-Lukas Tirpitz[1,a], Udo Frieß[1], Robert Spurr[2], and Ulrich Platt[1,3]

[1]Institute of Environmental Physics, University of Heidelberg, Heidelberg, Germany
[2]RT Solutions Inc., Cambridge, MA, USA
[3]Max Planck Institute for Chemistry, Mainz, Germany
[a]now at: Airyx GmbH, Justus-von-Liebig-Straße 14, 69214 Eppelheim, Germany

**Correspondence:** Jan-Lukas Tirpitz (jan-lukas.tirpitz@iup.uni-heidelberg.de)

**Abstract.** Ground-based Multi-AXis Differential Optical Absorption Spectroscopy (MAX-DOAS) is a widely-used measurement technique for the remote detection of atmospheric aerosol and trace gases. The technique relies on the analysis ultra-violet and visible radiation spectra of scattered sunlight (skylight) to obtain information on different atmospheric parameters. From an appropriate set of spectra recorded under different viewing directions (typically a group of observations at different elevation

angles) the retrieval of aerosol and trace gas vertical distributions is achieved through numerical inversion methods.

It is well known that the polarisation state of skylight is particularly sensitive to atmospheric aerosol content as well as aerosol properties, and that polarimetric measurement could therefore provide additional information for MAX-DOAS profile retrievals; however such measurement have not yet been used for this purpose. To address this issue, we have developed the RAPSODI (Retrieval of Atmospheric Parameters from Spectroscopic Observations using DOAS Instruments) algorithm.

In contrast to existing MAX-DOAS algorithms, it can process polarimetric information, and it can retrieve simultaneously profiles of aerosols and various trace gases at multiple wavelengths in a single retrieval step; a further advantage is that it contains a Mie scattering model, allowing for the retrieval aerosol microphysical properties. The forward model component in RAPSODI is based on a linearized vector radiative transfer model with Jacobian facilities, and we have used this model to create a data base of synthetic measurements in order to carry out sensitivity analyses aimed at assessing the potential of polarimetric

MAX-DOAS observations. We find that multispectral polarimetry significantly enhances the sensitivity, particularly to aerosol related quantities. Assuming typical viewing geometries, the degree of freedom for signal (DFS) increases by about $50\%$ and $70\%$ for aerosol vertical distributions and aerosol properties, respectively, and by approximately $10\%$ for trace gas vertical profiles. For an idealised scenario with a horizontally homogeneous atmosphere, our findings predict an improvement in the inversions results' accuracy (root-mean-square deviations to the true values) of about $60\%$ for aerosol VCDs as well as for

aerosol surface concentrations, and by $40\%$ for aerosol properties. For trace gas VCDs, very little improvement is apparent, although the accuracy of trace gas surface concentrations improves by about $50\%$.



## 1 Introduction

Multi-AXis Differential Optical Absorption Spectroscopy (MAX-DOAS) (e.g. Hönninger and Platt, 2002; Hönninger et al.,
2004; Wagner et al., 2004; Heckel et al., 2005; Frieß et al., 2006; Platt and Stutz, 2008; Irie et al., 2008; Clémer et al., 2010;
Wagner et al., 2011; Vlemmix et al., 2015) is a versatile passive remote sensing technique for the simultaneous detection
of aerosol and trace gases. The typical MAX-DOAS instrument (see also Section 2) consists of a motorised telescope and a
spectrometer unit, measuring ultraviolet (UV)- and visible (Vis) radiation spectra of scattered sunlight in different viewing
directions (Multi-Axis). The spectra are analysed using Differential Optical Absorption Spectroscopy (DOAS, Platt and Stutz,
2008), to obtain information on different atmospheric parameters, integrated over the effective light path from the top of
the atmosphere (TOA) to the instrument. From these spectra, one can retrieve tropospheric aerosol and trace gas vertical
distributions as well as aerosol properties by applying inverse modelling approaches, using dedicated retrieval algorithms.
Throughout this study we focus on the ground-based application of MAX-DOAS. It is widely used, not least because it allows
us to measure the atmospheric state with modest financial and logistical effort and with relatively simple instrumentation.

Due to limitations in instrument design and retrieval algorithm applicability, MAX-DOAS measurements at present do not
exploit the full information contained in skylight observations. For instance, information from broad-band spectral features,
inelastic scattering and saturation effects is either ignored or at best partially included in retrievals. Most important for this
study, however, is the information contained in the light's polarisation state and its spectral dependence. It has largely been
ignored (apart from few investigations e.g. by Seidler (2008) and Bernal (2017)), even though it is well known that its con-
sideration enhances the sensitivity of passive remote sensing to aerosol abundances and properties (e.g. Herman et al., 1971;
Mishchenko and Travis, 1997; Boesche et al., 2006; Hasekamp and Landgraf, 2007; Hansen and Travis, 1974). Indeed, po-
larimetry has been extensively utilized in recent decades for aerosol detection with radiometers/polarimeters, most prominently
in the "Aerosol Robotic Network" (AERONET Holben et al., 1998; Li et al., 2009), and the "POLarization and Directionality of
Earth's Reflectances" (POLDER) satellite instrument (Deschamps et al., 1994). Such polarimeters derive aerosol abundances
and properties from measurements of the polarisation state and absolute radiance of direct and scattered solar radiation at low
spectral resolution (applying narrow-band optical bandpass filters at few individual wavelengths, typically covering a spectral
range from approximately $400$ to $1000\,\mathrm{nm}$).

Despite these advances within the radiometer community, the application of polarimetry for MAX-DOAS measurements
comes with fundamentally new aspects, challenges and possibilities, arising from crucial differences between the MAX-DOAS
technique and radiometer observations. Firstly, MAX-DOAS instruments are typically not radiometrically calibrated and there-
fore do not provide absolute radiances. Instead, observations performed at different viewing directions are evaluated against
each other to obtain differential quantities (see also Section 2.2). Compared to radiometers, this process significantly simplifies
the instrumental setup and calibration, and improves long-term stability; however, it comes with a certain loss of informa-
tion from the measurements. Secondly, MAX-DOAS focusses on the detection of atmospheric trace gases with characteristic
narrow-band absorption features that require much higher spectral resolution ($\approx 1\,\mathrm{nm}$, see also Section 2.1) than the coarser
spectral bands typical of radiometers. Information on atmospheric aerosols is primarily required to confine the radiative trans-





port. It is not inferred from radiance measurements, but indirectly by analysing narrow-band absorption features of proxy gases with well known vertical distributions (e.g. the oxygen collision complex $O_4$, see also Section 2.3 Wagner et al., 2004; Frieß et al., 2006).

The major aim of this study is to assess the potential of polarimetry for the retrieval of trace gas and aerosol vertical distributions as well as aerosol properties from ground-based MAX-DOAS observations. For this purpose we have developed a novel MAX-DOAS inversion algorithm called RAPSODI (Retrieval of Atmospheric Parameters from Spectroscopic Observations using DOAS Instruments) with the capability to process polarimetric information. Our investigations are based on synthetic data: using the RAPSODI forward model (see Section 5.3), measurements are simulated for a wide range of atmospheric

conditions, measurement scenarios and retrieval settings. The simulated measurements are then fed back to the retrieval to investigate how well the original atmospheric state can be reproduced.

    Using synthetic data instead of field measurements for the assessment of the retrieval's performance has several advantages (Frieß et al., 2006, 2019): firstly, the validation of real MAX-DOAS measurements is extremely difficult and cumbersome since representative and accurate independent observations for the entire set of retrieved atmospheric parameters are required. Sec-

ondly, measurements need to be performed over long time periods to sample a wide range of atmospheric conditions, thereby covering the entire parameter space of interest. In contrast, when using synthetic measurements, the underlying atmospheric conditions can be arbitrarily chosen and are perfectly known beforehand; this allows us to carry out a controlled and systematic analysis.

    The paper is structured as follows: Section 2 describes the general MAX-DOAS technique as it is typically applied. Section

3 and Section 4 describe and motivate the major modifications in the instrumental setup, the measurement procedure and the inversion algorithm that are required to extend conventional MAX-DOAS by polarimetry. Section 5 describes the RAPSODI retrieval algorithm. Strategy, setup and results of our investigations are presented in Sections 6 to 11 and summarised in Section 12.

## 2   The MAX-DOAS technique

The procedure to determine the atmospheric state from MAX-DOAS observations can be subdivided into three major steps, to be discussed in the following subsections: 1) the actual measurement process (acquisition of skylight spectra); 2) the DOAS spectral analysis, which enables us to derive suitable intermediate quantities from the raw spectra; and 3) based on these intermediate results, the inversion procedure to determine the desired atmospheric parameters. The synthetic data used in this study comprises groups of directly simulated DOAS spectral analysis results. Therefore, steps 1) and 2) are not of critical

importance in this study; however, they will be described briefly in order to provide a complete picture of the method.

### 2.1   Spectra acquisition

Currently, a large number of different MAX-DOAS instrument prototypes are in existence; all are different in detailed implementation and sometimes they are optimised for special purposes. However, most of them share essential properties that have





been established over recent years (see e.g. Kreher et al., 2019) and will be outlined here. The typical MAX-DOAS instrument consists of a telescope and a grating spectrometer unit. The telescope features a narrow field of view (FOV), with full aperture angles of approximately $0.5°$ ($10\,\mathrm{mrad}$). It is motorised to realise multiple viewing directions in rather quick succession ($\approx 1\,\mathrm{min}$ per viewing direction). Telescopes are either installed at a fixed viewing azimuth angle (1D-MAX-DOAS), automatically changing the viewing elevation angle, or they have motors for both azimuth and elevation angles (2D-MAX-DOAS). There are different approaches to guide the gathered light from the telescope to the spectrometer, the most common being to use fused silica optical fibres. Typical spectral coverages and resolving powers of the spectrometer unit are on the order of $150\,\mathrm{nm}$ and $10^3$ (resolution of $\approx 1\,\mathrm{nm}$), respectively. It is not unusual to apply multiple spectrometers, commonly two, dedicated to UV and Vis spectral ranges with approximate coverages of $300 - 400\,\mathrm{nm}$ and $400 - 500\,\mathrm{nm}$, respectively.

In the following we will consider spectra $I(\lambda, \boldsymbol{\Omega})$. Here, $I$ is the radiance at wavelength $\lambda$, detected at the viewing geometry configuration $\boldsymbol{\Omega} = \{\theta, \phi, \alpha\}$, with $\theta$, $\phi$, $\alpha$ the solar zenith angle (SZA), relative azimuth angle (RAA) and viewing elevation angle (EA), respectively. A typical dataset required for a single MAX-DOAS retrieval consists of five to ten spectra recorded for different geometrical configurations. It is most common to perform so-called elevation scans, in which spectra are recorded at a fixed value of RAA for a range of EAs between a few degrees and the zenith.

## 2.2 DOAS spectral analysis

The first processing step in a MAX-DOAS evaluation is the DOAS analysis of the observed skylight spectra $I(\lambda, \boldsymbol{\Omega})$. $I(\lambda, \boldsymbol{\Omega})$ corresponds to the solar spectrum $I_{\mathrm{toa}}(\lambda)$ at TOA, altered by extinction of the light on molecules, particles and the Earth's surface on its way (actually the superpostion of a multitude of ways) through the atmosphere to the instrument. The observed slant optical thickness (SOT) is defined as

$$\tau(\lambda, \boldsymbol{\Omega}) = \ln\left(\frac{I_{\mathrm{toa}}(\lambda)}{I(\lambda, \boldsymbol{\Omega})}\right) \tag{1}$$

To a good approximation, $\tau(\lambda, \boldsymbol{\Omega})$ can be expressed according to the well-known DOAS model

$$\tau(\lambda) = \sum_s \sigma_s(\lambda)\, S_s + \sum_i^N b_i \lambda^i + k R(\lambda) + C(\lambda). \tag{2}$$

For readability, we have omitted the $\boldsymbol{\Omega}$ symbol here. The first term represents absorption by a series of trace gases, with absorption cross sections $\sigma_s(\lambda)$ and slant column densities (SCD) $S_s$ given by

$$S_s = \int_0^L c_s\, dl \tag{3}$$

Here, $c_s$ is the trace gas concentration integrated along the effective light path $L$ through the atmosphere. The second term in Eq. 2 is a polynomial ($N \approx 5$) with coefficients $b_i$ that accounts for spectral broadband features caused by scattering (into as well as out of the light path), aerosol absorption and reflectance from the Earth's surface. $R(\lambda)$ is the Raman optical depth with amplitude $k$. This represents those narrowband features caused by the filling-in of Fraunhofer lines and trace gas





absorption signatures due to inelastic rotational-Raman scattering by air molecules (Grainger and Ring, 1962; Solomon et al., 1987; Bussemer, 1993; Wagner, 1999). $C(\lambda)$ represents other potentially non-linear correction terms (arising for example from spectral shifts or from limited instrumental resolution); this term is not really relevant for our purposes and we will not consider it further in this work. For the detection of most trace gases, differences in optical depth smaller than $10^{-3}$ have to be detected reliably. Due to instrumental imperfections, this is typically not possible when using independent solar spectra $I_{\mathrm{toa}}(\lambda)$ from the literature in Eq. 1. Instead, SOTs are compared against spectra from the same instrument but at another viewing geometry $\mathbf{\Omega}_0$, yielding the so-called *differential* slant optical thicknesses (dSOT)

$$\Delta\tau(\lambda, \mathbf{\Omega}, \mathbf{\Omega}_0) = \ln\left(\frac{I(\lambda, \mathbf{\Omega}_0)}{I(\lambda, \mathbf{\Omega})}\right) \tag{4}$$

$$= \sum_s \sigma_s(\lambda)\, \Delta S_s(\mathbf{\Omega}, \mathbf{\Omega}_0) + \sum_i b_i \lambda^i + k R(\lambda) \tag{5}$$

This is equivalent to the formulation in Eq. 2, but with the SCD $S_s$ replaced by the *differential* SCD (dSCD) $\Delta S_s$:

$$\Delta S_s(\mathbf{\Omega}, \mathbf{\Omega}_0) = S_s(\mathbf{\Omega}) - S_s(\mathbf{\Omega}_0). \tag{6}$$

Taking reference data from the literature for $\sigma_s(\lambda)$, the modelled dSOT in Eq. 5 can be fitted to the observed dSOT in Eq. 4 using least-squares methods, with the fitting parameters being $\Delta S_s$, $b_i$ and $k$.

The set of dSCDs $\Delta S_s$ for the different trace gases constitutes the main output of the DOAS analysis, and this output is typically the sole input for conventional MAX-DOAS inversion algorithms. Indeed, in most applications the broadband information contained in the parameter set $b_i$ is discarded. In this study we also ignore $b_i$ at this point, and instead re-incorporate the broadband information into the inversion in terms of dSOTs at discrete wavelengths (see Section 5). The information inherent in the Ring-effect scaling parameter $k$ will be ignored in this study, even though it has been shown to provide additional information on aerosols (Wagner et al., 2009).

### 2.3 Inversion procedure

Inferring the state of the atmosphere (SOA) from the measured quantities (as introduced in Section 2.2) represents a non-linear inverse problem. The SOA is characterised by a number of parameters, such as the aerosol and trace gas vertical concentrations, aerosol optical properties and Earth surface reflection. In most cases, only a subset of these parameters is actually retrieved (see also Section 5.4).

Retrieval algorithms for the SOA make use of a radiative transport model (RTM) to reproduce the measurements (typically the dSCDs obtained from the DOAS spectral analysis) given a model atmosphere input. During the inversion, the model parameters are iteratively adapted to bring simulated and real measurements into closure, thereby approaching the real atmospheric state. In recent years, a multitude of MAX-DOAS inversion algorithms have been implemented that apply different approaches in terms of parameterisation, a priori constraints and optimisation schemes (Irie et al., 2008; Clémer et al., 2010; Wang et al., 2013; Yilmaz, 2012; Bösch et al., 2018; Chan et al., 2019; Vlemmix et al., 2011; Beirle et al., 2019).

In these retrievals, information on aerosols is typically inferred from the oxygen collision complex ($O_2$-$O_2$, in the following referred to as $O_4$). Its atmospheric concentration is proportional to the square of the $O_2$ concentration, and thus, its vertical





distribution is well known. The depth of $O_4$ absorption features in a skylight spectrum therefore provides information on the effective light path through the atmosphere. The latter is mostly driven by the abundance and properties of aerosol. Hence, accurate detection of $O_4$ absorption features not only allows us to fine-tune the radiative transfer but also enables us to retrieve aerosol vertical distributions and properties (Wagner et al., 2004; Frieß et al., 2006).

## 3 From conventional to polarimetric MAX-DOAS

In this Section we outline briefly the major modifications (on instrumental aspects as well as data evaluation) necessary to incorporate polarimetric information into the MAX-DOAS measurement procedure.

Regarding the implementation of a Polarimetric MAX-DOAS (PMAX-DOAS) instrument, various instrumental setups are conceivable. In this study we assume a PMAX-DOAS instrument that features a motorised linear polariser inside the telescope; this will record spectra of scattered skylight at arbitrary polarisation orientations or polariser angles (PA) $\delta$. For such spectra

(in the following referred to as "polarimetric spectra"), the viewing geometry configuration $\mathbf{\Omega}$ introduced in Section 2.2 is extended by $\delta$, hence $\mathbf{\Omega} = (\theta, \phi, \alpha, \delta)$. Throughout this study, the PA describes the orientation of the polariser's transmitting axis with respect to the horizon, increasing clockwise when looking towards the instrument. In the atmosphere, the contribution of circularly polarised light is negligibly small and usually ignored (Hansen, 1971). Information on skylight radiant intensity and state of polarisation (SOP) can then be fully captured by performing three radiance measurements $I_\delta$ at three different PAs,

ideally in steps of $\Delta\delta = 60°$ (see e.g. Xu and Wang, 2015). In this study, we assume for the most part that our PMAX-DOAS instrument records three spectra at PAs of $\delta \in \{0, 60, 120°\}$ in each viewing direction.

The DOAS spectral analysis is readily applicable to polarimetric spectra in the same manner as that described in Section 2.2, yielding what in the following will be referred to as "polarimetric dSCDs". Note, however, that the information on the SOP of skylight needs to be incorporated into the inversion to exploit the full potential of polarimetry. This information is

170 not contained in the polarimetric dSCDs themselves, but rather in the intensity ratios between spectra recorded at different $\delta$. Therefore, in addition to the polarimetric dSCDs, also "polarimetric dSOTs" (between polarimetric spectra, according to Eq. 4) at discrete wavelengths will be provided as additional input to the inversion algorithm. This issue is discussed in more detail in Section 5.2.

The major requirement for the retrieval algorithm is a forward model that is able to simulate both polarimetric dSCDs and

175 polarimetric dSOTs by means of a vectorised radiative transfer model (RTM). In this study, we use the vector discrete ordinate model VLIDORT (Spurr, 2006, 2008, 2021). Forward-model aspects are discussed in more detail in the description of the RAPSODI algorithm Section 5.2.

The SOP of skylight can be characterised in terms of the Stokes parameters $I$, $Q$, $U$ and $V$ (Stokes, 1851; Chandrasekhar,

2013). Here, $I$ (denoting the total radiance), $Q$ (radiance difference between electromagnetic wave components parallel and perpendicular to the reference plane) and $U$ (radiance difference between electromagnetic wave components at $45°$ and $135°$ to the reference plane) carry information on the state of linear polarisation, while parameter $V$ is linked to circular polarisation.





These Stokes parameters are related to our measurements according to:

$$I = \frac{2}{3}(I_{0°} + I_{60°} + I_{120°}) \tag{7}$$

$$Q = \frac{2}{3}(2I_{0°} - I_{60°} - I_{120°}) \tag{8}$$

$$U = \frac{2}{\sqrt{3}}(I_{60°} - I_{120°}) \tag{9}$$

$$V = 0 \tag{10}$$

Further useful quantities are the degree of linear polarisation (DOLP)

$$D = \frac{\sqrt{Q^2 + U^2}}{I} \tag{11}$$

$$= 2\frac{\sqrt{I_{0°}^2 + I_{60°}^2 + I_{120°}^2 - I_{0°}I_{60°} - I_{0°}I_{120°} - I_{60°}I_{120°}}}{I_{0°} + I_{60°} + I_{120°}} \tag{12}$$

and the angle of polarisation (AOP) $\chi$

$$\chi = \frac{1}{2}\arctan\left(\frac{U}{Q}\right) + \begin{cases} 90°, & \text{if } Q \geq 0 \\ 0°, & \text{if } Q < 0 \end{cases} \tag{13}$$

$$= \frac{1}{2}\arctan\left(\sqrt{3}\frac{I_{60°} - I_{120°}}{2I_{0°} - I_{60°} - I_{120°}}\right) + \begin{cases} 90°, & \text{if } Q \geq 0 \\ 0°, & \text{if } Q < 0 \end{cases}, \tag{14}$$

which corresponds to the polariser angle $\delta$ at which the observed intensity is maximised. However, in the present work, Equations 7 to 14 will play only a minor role, since the RAPSODI forward model directly simulates dSCDs and dSOTs (see Section 5.3) and a transformation between measurements $I_\delta$ and Stokes parameters is therefore not necessary for the inversion.

## 4 Remarks on skylight polarisation

In this section, we outline briefly the reasons why the application of polarimetry is expected to improve MAX-DOAS atmospheric state retrievals. Sunlight at TOA is initially unpolarised but becomes partially polarised through scattering processes in the Earth's atmosphere. Pioneering work in this direction dates back to the 19th century, notably with the discovery of skylight polarisation and its basic properties by François Arago in 1809 (Arago, 1862), as well as the first theoretical explanation by Strutt (1871). Different scatterers (molecules as well as aerosols) have very different effects on the SOP. Hence, the observed SOP of skylight, including its angular distribution and broadband spectral patterns, depends strongly on atmospheric aerosol abundances and properties, and can therefore be expected to provide significant amounts of additional information in retrievals. In recent decades, extensive investigations on the use of polarization measurements have been performed e.g. by Herman et al. (1971); Hansen and Travis (1974); Mishchenko and Travis (1997); Boesche et al. (2006); Dubovik et al. (2006); Hasekamp and Landgraf (2007); Li et al. (2009); Emde et al. (2010). In contrast, very little attention has been paid to the fact that the





light fields of different polarisation orientations arriving at the Earth's surface have taken different effective paths through the atmosphere. Polarimetric measurements therefore allow us to access new light path geometries; this is of particular importance for applications such as MAX-DOAS, which aim at the retrieval of spatial distributions of atmospheric species. This effect is
reflected by variation of airmass factors and box airmass factors with polariser angle $\delta$ as explained and illustrated in Supplement S1. In a Rayleigh atmosphere (no aerosols), variations of the $O_4$ SCD by up to $60\%$ in a single viewing direction can be achieved just by varying $\delta$. Even though SCD variation is lower for trace gases at low altitudes and in scenarios with enhanced aerosol loading, this polarisation effect is expected to improve the retrieval of vertical profiles for any species.

## 5 The RAPSODI retrieval algorithm

As mentioned in Section 1, the basis for our studies is the RAPSODI retrieval algorithm, which has been developed to overcome some of the current limitations of the above (Section 2.3) described MAX-DOAS retrieval algorithms. RAPSODI's most notable features are:

1. It is the first MAX-DOAS retrieval algorithm with the ability to account for the polarisation state of skylight, which is the key feature for the presented study.

2. It is the first algorithm to retrieve simultaneously all species of interest (aerosols and trace gases) from observations at multiple wavelengths (multispectral) in a single inverse-model step. Former algorithms performed separate inversions for aerosols (from the $O_4$ dSCDs) and each trace gas.

3. It is the first algorithm to retrieve information on aerosol microphysical properties (size distribution parameters and complex refractive indices) by making use of a Mie model. Formerly, information on aerosol properties was either
prescribed or derived in terms of simplified optical parameters (for example, the asymmetry parameter characterising the Henyey-Greenstein phase function model).

RAPSODI is fully backward-compatible, meaning that non-polarimetric monochromatic retrievals performed separately for aerosol and trace gases at a single wavelength are still possible, and Mie model output can be replaced by the Henyey-Greenstein approximation by setting corresponding flags in the retrieval configuration. Figure 1 represents a schematic overview
of the algorithm. The crucial components and functionalities are described in the following subsections. For a detailed description of the algorithm, the reader is referred to Tirpitz (2021).

### 5.1 Optimal estimation formalism

As with earlier MAX-DOAS algorithms, RAPSODI is based on the optimal estimation (OEM) inversion formalism described in detail in Rodgers (2000). The aim of inverse model is to find that SOA (described by a set of parameters summarised in
the state vector $\mathbf{x}$) which best reproduces a set of real observations $\hat{\mathbf{y}} = \mathbf{F}(\mathbf{x})$, given a forward model $\mathbf{F}$. This is achieved by





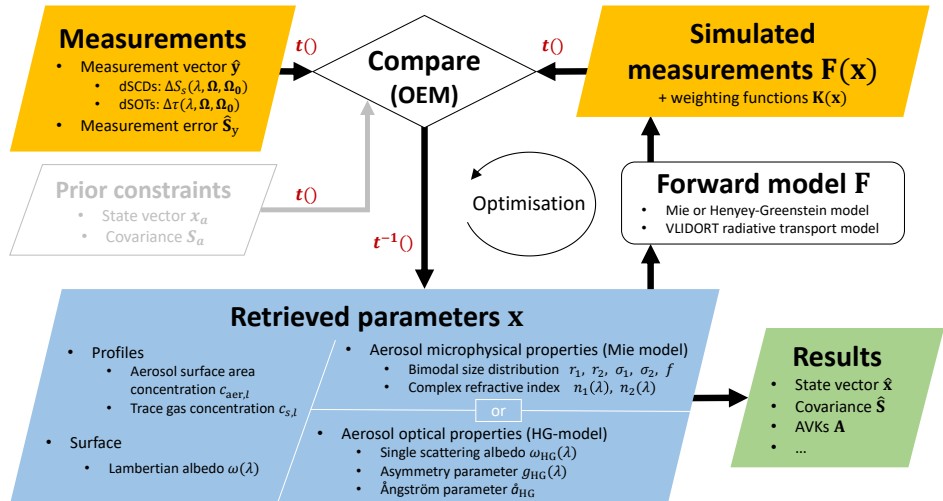

**Figure 1.** Schematic overview of the RAPSODI retrieval algorithm. The individual components are discussed in detail in the following subsections. Red labels next to arrows indicate transformations to/from optimised units, as described in Section 5.4.

minimizing the cost function

$$\chi^2 = (\hat{\mathbf{y}} - \mathbf{F}(\mathbf{x}))^T \mathbf{S}_y^{-1}(\hat{\mathbf{y}} - \mathbf{F}(\mathbf{x})) + (\mathbf{x} - \mathbf{x_a})^T \mathbf{S}_a^{-1}(\mathbf{x} - \mathbf{x_a}). \tag{15}$$

Here, $\mathbf{x}_a$ is the *a priori* state vector. $\mathbf{S}_y$ and $\mathbf{S}_a$ are the measurement and *a priori* covariance matrices. $\mathbf{x}_a$ and $\mathbf{S}_a$ describe estimates of the most likely atmospheric state and its variability prior to the inversion. Ideally, their values are inferred from climatologies or other independent datasets. The first and second terms in Eq. 15 tend to counterbalance each other, and this aspect is crucial to the understanding of some of the approaches taken in this study: the first term dominates for those state vector elements $x$ with high measurement sensitivity. In contrast, if the sensitivity of the measurements to $x$ is low, the second term dominates and $x$ is drawn towards its *a priori* value $x_a$. The minimisation problem is non-linear and the solution is found iteratively by applying a Levenberg-Marquardt optimisation scheme:

$$\mathbf{x}_{i+1} = \mathbf{x}_i + (\mathbf{K}_i^T \mathbf{S}_y^{-1} \mathbf{K}_i + (1+\gamma)\mathbf{S}_a^{-1})^{-1}$$
$$\times \left[ \mathbf{K}_i^T \mathbf{S}_y^{-1}(\hat{\mathbf{y}} - \mathbf{F}(\mathbf{x}_i)) - \mathbf{S}_a^{-1}(\mathbf{x}_i - \mathbf{x}_a) \right]. \tag{16}$$

The index $i$ indicates the current iteration, and $\mathbf{K} = \partial \mathbf{F}(\mathbf{x})/\partial \mathbf{x}$ is the weighting function matrix, representing the linearisation of the forward model for a distinct atmospheric state $\mathbf{x}$. Once the solution $\hat{\mathbf{x}}$ has been found, its covariance is calculated according to

$$\hat{\mathbf{S}} = (\mathbf{K}^T \mathbf{S_y}^{-1} \mathbf{K} + \mathbf{S_a}^{-1})^{-1} \tag{17}$$





A useful measure for the amount of information obtained on the different state vector elements is the averaging kernel (AVK) matrix, defined as

$$\mathbf{A} = \frac{\partial \hat{\mathbf{x}}}{\partial \mathbf{x}} = (\mathbf{K}^T \mathbf{S_y}^{-1} \mathbf{K} + \mathbf{S_a}^{-1})^{-1} \mathbf{K}^T \mathbf{S_y}^{-1} \mathbf{K} \tag{18}$$

The diagonal elements $A_{jj}$ lie between zero and unity for each state vector element $x_j$; they indicate the relative amount of information on $x_j$ gleaned from the measurements in relation to that from *a priori* knowledge. The trace of $\mathbf{A}$ is often referred

to as the degrees of freedom for signal (DOFS), a quantity that will play a crucial role in this study. Note that, in the following, we will sometimes refer by "DOFS" to the sum over only a few values (or even just a single value) of the AVK diagonal. The off-diagonal entries $A_{jk}$ ($j \neq k$) contain information on the cross-sensitivity of $x_j$ to all other parameters $x_k$. In the ideal case of a retrieval achieving full sensitivity with respect to all state vector elements and no cross-correlations, $\mathbf{A} = \mathbf{1}$ (the identity matrix).

## 5.2 Measurement vector

The RAPSODI measurement vector $\mathbf{y}$ consists of dSCDs $\Delta S_s(\lambda, \mathbf{\Omega}, \mathbf{\Omega}_0)$ and dSOTs $\Delta \tau(\lambda, \mathbf{\Omega}, \mathbf{\Omega}_0)$ for different trace gas species $s$ (including $O_4$), viewing geometry $\mathbf{\Omega}$, reference geometry $\mathbf{\Omega}_0$ and radiation wavelength $\lambda$. With conventional non-polarimetric dSOTs and dSCDs, $\mathbf{\Omega}$ and $\mathbf{\Omega}_0$ comprise just the viewing direction ($\theta$, $\phi$ and $\alpha$), whereas with polarimetric dSOTs and dSCDs, these configurations also contain the polariser angle $\delta$.

DSCDs carry information on trace gas abundances and, in the case of $O_4$, on the effective light path. Most conventional MAX-DOAS retrieval algorithms accept dSCDs as the only input. RAPSODI also accepts dSOTs, to allow for the incorporation of two further kinds of information: 1) dSOTs between spectra recorded in the same viewing direction ($\theta$, $\phi$, $\alpha$) but at different PAs $\delta$ contain information on the light's SOP and - if dSOTs are provided at multiple wavelengths - the SOP's spectral dependence. 2) In contrast, dSOTs between spectra of different viewing directions but with same $\delta$ contain information on the

spectral broadband variation of the radiance over the sky hemisphere.

## 5.3 Forward model

The RAPSODI forward model is based on VLIDORT (Spurr, 2006, 2008, 2021), a pseudo-spherical 1-D vector discrete ordinate RTM. VLIDORT has the ability to generate fields of analytically derived weighting functions (Jacobians) with respect to any atmospheric and/or surface properties. VLIDORT is coupled with a Mie scattering model (De Rooij and Van der

Stap, 1984; Spurr et al., 2012) that calculates the aerosol optical properties of relevance for the RTM from a set of aerosol microphysical properties; the Mie model is also analytically differentiable with respect to the aerosol microphysical parameter inputs. Thus, the forward model not only simulates dSCDs and dSOTs but it also generates the corresponding weighting functions $\mathbf{K}$.

For the Mie model we have assumed a bimodal log-normal size distribution as described in more detail in Section 5.4. The

Mie scattering model can be replaced by the Henyey-Greenstein (HG) approximate phase function by setting a corresponding flag in the retrieval settings.





## 5.4 State vector

Elements of the state vector $\mathbf{x}$ are the desired retrieval parameters. These include the concentrations $c_{s,l}$ of species $s$ (aerosols and various trace gases, $O_4$ excluded) in each model layer $l$, the Lambertian surface albedo $\omega_{\text{surf}}(\lambda)$ and – depending on the

aerosol model used – optical or microphysical aerosol properties. A $\lambda$-dependency indicates that the corresponding parameter is treated as spectrally resolved, meaning that separate values are retrieved at each wavelength for which measurements (either dSCDs or dSOTs) are present. If the Henyey-Greenstein approximation is in force, the asymmetry parameters $g_{\text{hg},l}(\lambda)$, the single scattering albedos $\omega_{\text{hg},l}(\lambda)$ and the Ångström parameter $\mathring{a}_{\text{hg},l}$ appear in the state vector. If the Mie model is used, the state vector contains the median radii $r_{l,m}$ of each mode $m \in \{1,2\}$, the modal widths $\sigma_{l,m}$, the modal fraction $f_l$ and the

complex refractive indices $n_{l,m}(\lambda)$. All aerosol parameters can be linked over $\lambda$, $l$ or $m$, thereby allowing us to retrieve average values over altitude, wavelength or size distribution modes.

For the present study, we will make use of the Mie-model as the HG-model cannot reproduce a realistic state of skylight polarisation (Tirpitz, 2021) and therefore does not represent a suitable approximation for polarimetric retrievals. Furthermore, we will assume the same aerosol properties for all model layers; thus we omit the layer index $l$ in the following discussion.

Hence, in this study aerosol property parameters contained in state vector are the median radii $r_m$ of each mode $m \in \{1,2\}$, the modal widths $\sigma_m$, the modal fraction $f$ and the complex refractive index $n_m$. $r_m$, $\sigma_m$ and $f$ parametrise the bimodal log-normal aerosol size distribution according to

$$\frac{\mathrm{d}N(r)}{\mathrm{d}r} = \frac{1}{\sqrt{2\pi}r} \left[ \frac{f}{\sigma_1} \exp\left( -\frac{1}{2} \left( \frac{\ln(r/r_1)}{\sigma_1} \right)^2 \right) \right.$$
$$\left. + \frac{1-f}{\sigma_l} \exp\left( -\frac{1}{2} \left( \frac{\ln(r/r_2)}{\sigma_2} \right)^2 \right) \right] \tag{19}$$

with $\frac{\mathrm{d}N}{\mathrm{d}r}$ being the normalised number of particles $N$ per radius interval.

Conventional algorithms retrieve the aerosol abundances in terms of an extinction coefficient $\mu_{\text{aer}}$ that generally depends on $\lambda$; this is not convenient for multispectral retrievals, where a single aerosol profile shall be inferred from observations at multiple wavelengths. RAPSODI therefore quantifies aerosol in terms of the $\lambda$-independent surface area concentration $c_{\text{aer}}$ in units of $[\mu\text{m}^2\,\text{cm}^{-3}]$, and so vertical columns $V_{\text{aer}}$ are not expressed in terms of the familiar aerosol optical thickness (AOT) but instead in units of $[\mu\text{m}^2\,\text{cm}^{-2}]$. Also the results of this study are presented in these units. Note that $\mu_{\text{aer}}$ and AOT are related to

$c_{\text{aer}}$ and $V_{\text{aer}}$ according to

$$\mu_{\text{aer}}(\lambda) = \frac{1}{4} E(\lambda)\, c_{\text{aer}} \tag{20}$$

$$\text{AOT}(\lambda) = \frac{1}{4} E(\lambda)\, V_{\text{aer}}. \tag{21}$$

$$\tag{22}$$

with $E(\lambda)$ being the aerosol bulk extinction efficiency which depends on the aerosol properties. It can be determined by Mie

calculations and ranges between 0.5 and 3 for typical atmospheric aerosol in the UV-Vis spectral range.





**Table 1.** Overview of the simulated observations. Wavelengths are in nm.

| Observations | $\lambda$ | Absorption cross section | Assumed error |
|---|---|---|---|
| dSOTs | 343, 360, 415 | N.A. | 0.02 |
| | 460, 477, 532 | | |
| $O_4$ dSCDs | 360, 477 | 293 K, Thalman and Volkamer (2013) | $2 \cdot 10^{41} \, \text{molec}^2 \, \text{cm}^{-5}$ |
| HCHO dSCDs | 343 | 297 K, Meller and Moortgat (2000) | $2 \cdot 10^{15} \, \text{molec} \, \text{cm}^{-2}$ |
| $NO_2$ dSCDs | 360, 460 | 298 K, Vandaele et al. (1998) | $5 \cdot 10^{14} \, \text{molec} \, \text{cm}^{-2}$ |

As is the case with earlier OEM retrieval algorithms (Yilmaz, 2012; Friedrich et al., 2019; Wang et al., 2013), RAPSODI can transform individual state vector elements to numerically more favourable quantities $\mathbf{x}'$ before the OEM formalism is applied (in Figure 1, these transformations are indicated by "$t()$"). In this work, we make use of two kinds of transformations: (1) the "log"-transformation allowing us to retrieve parameters in logarithmic space, hence $x' = \ln(x)$. In this way, negative values in

$x$ are avoided and larger values of $x$ are allowed; (2) the "frac"-transformation maps fractional quantities speficied over the interval [0,1] (such as albedos, asymmetry parameters and modal fractions) into the $(-\infty, \infty)$ space through the transformation $x' = -\ln(1/x - 1)$, which is the inverse of a Fermi-Dirac distribution. The main motivations for these transformations are: (1) they avoid unphysical results and numerical failures due to discrete boundaries in the parameter values, and (2) the OE inversion formalism by Rodgers (2000) assumes parameter variations and uncertainties to be normally distributed, a condition which is

often better fulfilled for suitably transformed parameter variables. Transformations for all parameters are listed in Table 4.

## 6 Synthetic scenarios and RTM settings

The synthetic data set for our studies was created by simulating measurements using the RAPSODI forward model. Where possible, the atmospheric scenarios and settings have been adapted from Frieß et al. (2019), who performed a comparison of eight retrieval algorithms also on the basis of synthetic measurements.

Table 1 provides an overview of the simulated observations. Simulations are performed at six different wavelengths $\lambda \in \{343, 360, 415, 460, 477, 532 \, \text{nm}\}$. DSCDs are simulated for $O_4$, Formaldehyde (HCHO) and Nitrogen dioxide ($NO_2$). As shown in Table 1, dSOTs are simulated at all six wavelengths, whereas trace gas dSCDs are limited to those wavelengths where listed trace gases show significant OTs to be detected by DOAS. The dSCD simulation wavelengths are representative for typical DOAS spectral fitting ranges and were adapted from the settings used in Tirpitz et al. (2021). Assumed measurement

uncertainties for dSCDs were taken from Frieß et al. (2019), while the uncertainties for dSOTs are inspired by investigations carried out on field data in Tirpitz (2021).

Vertical profiles of aerosol and trace gases used for the simulation calculations are shown in Figure 2, and their key properties are listed in Table 2. Compared to Frieß et al. (2019), the number of different aerosol profiles is reduced in number: extreme cases with optically thick fog and cloud layers are not considered. Synthetic observations were simulated for all 63


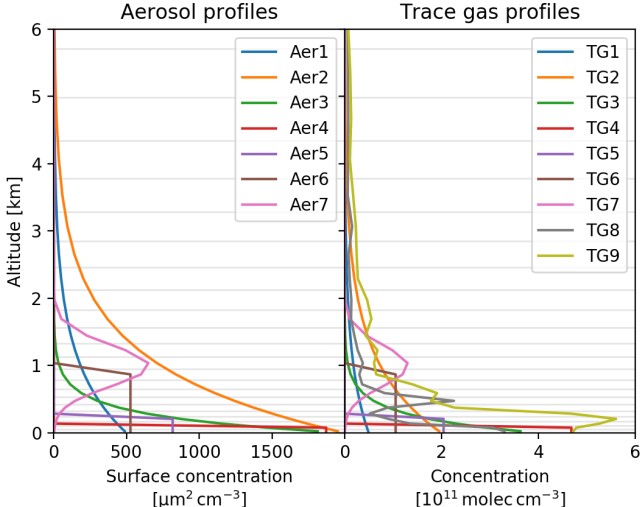

**Figure 2.** Aerosol area concentrations (left) and trace gas concentrations (right, same profiles for HCHO and NO$_2$) vertical profiles used as input for the forward modelling of synthetic data (see also Table 2)

.

combinations of aerosol and trace gas profiles, always applying a common trace gas profile for both NO$_2$ and HCHO. For pressure, temperature and humidity profiles, we assumed a US standard atmosphere. Ozone absorption was not included.

For the model atmosphere, we chose an exponential height grid (increasing layer thickness with altitude) with $l_{max} = 40$ layers extending from $0\,\mathrm{km}$ to $h_{l_{max}} = 60\,\mathrm{km}$ and a surface layer height of $50\,\mathrm{m}$. The layer boundary altitudes are given by

$$h_l = a(b^l - 1) \tag{23}$$

with $l \in [0, 40]$ and constants $a = 0.3676$ and $b = 1.13602$. The original regular-grid profiles used in Frieß et al. (2019) were interpolated to this grid, resulting in an occasional slight change of shape (however, total vertical columns were preserved). Even though 40 layers were considered for the forward model vertical discretization, only quantities in the lowest 25 layers between 0 and $\approx 8.5\,\mathrm{km}$ altitude were actually retrieved.

We performed simulations for the solar and instrument viewing geometries as listed in Table 3. Measurements without
polariser are indicated by a dash ($\delta = -$). In addition to the commonly performed elevation scans, we have included solar almucantar scans (RAA sequence at solar elevation); these are known to provide additional information on aerosol properties, in particular when the solar aureole region (viewing directions close to the sun) is sampled (Herman et al., 1971). The RAA values applied in this study were taken from Dubovik and King (2000). Observations were simulated for each combination of EA, SZA, RAA and PA. We assume the instrument to be located on the ground at sea level. In the following, we will refer
to the set comprising an aerosol profile, a trace gas profile, a distinct SZA and an elevation scan RAA as and "atmospheric scenario".


**Table 2.** Characterization of the aerosol and trace gas profiles depicted in Figure 2. Aerosol VCDs are in units of $10^8\,\mu m^2 cm^{-2}$, while trace gas VCDs have units $10^{15}\,molec\,cm^{-2}$

| Profile | Description | VCD |
|---|---|---|
| Aer0 | No aerosol | 0.0 |
| Aer1 | Exponential, 1 km scale height | 0.5 |
| Aer2 | Exponential, 1 km scale height | 2.0 |
| Aer3 | Exponential, 250 m scale height | 0.5 |
| Aer4 | Box profile, 107 m height | 0.2 |
| Aer5 | Box profile, 245 m height | 0.2 |
| Aer6 | Box profile, 948 m height | 0.5 |
| Aer7 | Gaussian at 1 km, 300 m FWHM | 0.5 |
| TG0 | No trace gas | 0.0 |
| TG1 | Exponential, 1 km scale height | 5.0 |
| TG2 | Exponential, 1 km scale height | 20.0 |
| TG3 | Exponential, 250 m scale height | 10.0 |
| TG4 | Box profile, 107 m height | 5.0 |
| TG5 | Box profile, 245 m height | 5.0 |
| TG6 | Box profile, 948 m height | 10.0 |
| TG7 | Box profile, 948 m height | 10.0 |
| TG8 | $NO_2$ balloon sonde profile[a] (2016-09-14) | 17.73 |
| TG9 | $NO_2$ balloon sonde profile[a] (2016-09-21) | 40.88 |

[a] Balloon sonde flights were performed during the CINDI-2 campaign in the
Netherlands in 2016 (Kreher et al., 2019)

Aerosol microphysical parameter settings are noted in Table 4. These represent mixed aerosols (mixture of oceanic and industrial aerosol) as reported by Dubovik et al. (2002) for the Maldives, and they are the same in all simulations. The effect of different aerosol sizes is discussed separately (see Figure 7). The size distributions, scattering matrix elements and bulk optical

properties are illustrated in Figures S3 and S4 in the supplement.

We have adopted *a priori* correlations for both spatially and spectrally resolved state vector elements. For vertical profiles, the correlation coefficient between concentrations at different altitudes decays exponentially with the vertical distance between the corresponding layers. Similarly, for $\omega_{\text{surf}}(\lambda)$, $n_1(\lambda)$ and $n_2(\lambda)$, correlations between state vector elements at different wavelengths decay with the spectral distance. Hence, off-diagonal elements of the sub-matrices of the *a priori* covariance $\mathbf{S}_a$

are given by

$$S_{a,ij} = \sqrt{S_{a,ii}\,S_{a,jj}}\exp\left(-\frac{\Delta l}{l_0}\right). \tag{24}$$




**Table 3.** Solar and instrument viewing geometries used for the simulation of synthetic observations. Simulations were performed for each combination of EA, SZA, RAA and PA. A dash in the PA indicates non-polarimetric observations.

| Parameter | Values [°] |
|---|---|
| SZA ($\theta$) | 40, 60, 90 |
| RAA ($\phi$) | Elevation scans: |
| | 0, 90, 180° |
| | Almucantar scan: |
| | 2, 2.5, 3, 3.5, 4, 5, 6, 10, 12, 14, 16, |
| | 18, 20, 25, 30, 35, 40, 45, 50, 60, 70, |
| | 80, 90, 100, 120, 140, 160, 180 |
| EA ($\alpha$) | 1, 2, 3, 4, 5, 6, 8, 15, 30, 90 |
| PA ($\delta$) | -, 0, 60, 120 |

Here, $l$ is the spatial or spectral distance, while $l_0$ defines the correlation length. Values of $l_0$ used in this work are listed in Table 4.

The resulting synthetic dataset consists of nearly one million dSOTs and dSCDs and about one hundred million associated
weighting functions, simulated for all possible combinations of aerosol profiles, trace gas profiles, and viewing geometries as noted above. Any subset of the simulated observations can be condensed into a synthetic measurement vector $\hat{y}$ to be ingested into the RAPSODI retrieval algorithm. In particular, observations can be added or removed, and the resulting effects on the inversion results can be examined systematically. We investigate various sets of measurements, in the following referred to as "measurement modes". $\hat{y}$ always includes a full elevation scan with ten EAs as listed in Table 3. Depending on the measurement
mode, the corresponding solar almucantar scan may be included. For convenience in the analysis, each measurement mode is labeled by a unique code, which consists of a series of several flags as defined in Table 5, these flags indicating which observations have been incorporated. A few examples are given here for illustration: The mode labels 'UV' and 'Vis' indicate conventional retrievals from non-polarimetric dSCDs for each species at a single wavelength in the UV and Vis spectral ranges respectively; for these modes we assume that all species are retrieved separately. In contrast, the mode label 'Multi-S-P-A'
indicates that the measurement set was extended to all wavelengths, that all species are retrieved simultaneously, and that polarimetric observations as well as an almucantar scan have been included.

It should be noted that, depending on the measurement mode, the vector $\hat{y}$ for a single retrieval can contain up to 1250 measurements (dSCDs and dSOTs for different species, wavelengths and elevation scan as well as solar almucantar scan geometries), while the state vector $\hat{x}$ can contain up to 110 parameters (profile concentrations of three different species in 25
layers, plus surface albedo and aerosol parameters that may be spectrally resolved).





**Table 4.** Settings for state vector elements **x** used in this study. The "true values"-column indicates the values applied for the forward simulation of synthetic observations. Values for surface albedo and aerosols were adapted from Dubovik et al. (2002). Typically observed values for trace gas concentrations were derived from data presented in Tirpitz et al. (2021). Remaining columns enumerate the *a priori* settings used for the inversion procedure.

| Parameter | Symbol | Transfor-mation | True value | Apriori value | Apriori uncert. | Corr. length | Typically observed |
|---|---|---|---|---|---|---|---|
| Aerosol area conc. | $c_{\mathrm{aer},l}$ | log | see Fig. 2 | Exp. profile[a] | 50 % | 1 km | $(6.6 \pm 5.6) \cdot 10^2 \, \mu m^2/cm^3$ |
| HCHO conc. | $c_{\mathrm{HCHO},l}$ | log | see Fig. 2 | Exp. profile[b] | 50 % | 1 km | $(5.0 \pm 3.4) \cdot 10^{10} \, molec/cm^3$ |
| NO₂ conc. | $c_{\mathrm{NO2},l}$ | log | see Fig. 2 | Exp. profile[c] | 50 % | 1 km | $(18 \pm 10) \cdot 10^{10} \, molec/cm^3$ |
| Surface albedo | $\omega_{\mathrm{surf}}$ | frac | 0.043 | 0.054 | 0.03 | 400 nm | $0.043 \pm 0.011$ [f] |
| Fine mode radius | $r_1$ | log | $0.095 \, \mu m$ | $0.111 \, \mu m$ | 30 % | - | $(0.095 \pm 0.016) \, \mu m$ |
| Coarse mode radius | $r_2$ | log | $0.49 \, \mu m$ | $0.43 \, \mu m$ | 30 % | - | $(0.49 \pm 0.06) \, \mu m$ |
| Fine mode width | $\sigma_1$ | log | 0.46 | 0.5 | 20 % | - | $0.46 \pm 0.04$ |
| Coarse mode width | $\sigma_2$ | log | 0.76 | 0.71 | 20 % | - | $0.76 \pm 0.05$ |
| Real refr index | $\Re n_1, \Re n_2$ | - | 1.44 | 1.46 | 0.1 | 400 nm | $1.44 \pm 0.02$ [f] |
| Imag. refr. index | $\Im n_1, \Im n_2$ | log | 0.011 | 0.018 | 100 % | 400 nm | $0.011 \pm 0.007$ [f] |
| Modal fraction | $f$ | frac | 0.9983 | 0.996 | 0.003 | - | $0.9983 \pm 0.0023$ |

[a] 1 km scale height, VCD of $0.5 \cdot 10^8 \, \mu m^2 \, cm^2$

[b] 1 km scale height, VCD of $8 \cdot 10^{15} \, molec/cm^2$

[c] 1 km scale height, VCD of $9 \cdot 10^{15} \, molec/cm^2$

[f] Value at 440 nm

# 7 Analysis of the measurements' information content

A major aim of this study is to investigate how the information content on the state vector elements **x** depends on different measurement modes, particularly with the addition of polarimetric observations. For the quantification of information, we employ the degrees of freedom for signal (DOFS) concept, as introduced in Section 5.1. Note that, for the calculation of DOFS, we do not need to perform an actual iteratative OEM inversion, since the averaging kernel in Eq. 18 requires knowledge of the a priori covariance matrix $\mathbf{S}_a$, the measurement covariance matrix $\mathbf{S}_y$, in addition to the weighting function matrix $\mathbf{K}$, calculated for a distinct atmospheric state $\hat{\mathbf{x}}$. These weighting functions have been calculated already as part of the creation of the synthetic observation dataset. It is now straightforward to derive DOFS for any atmospheric scenario, measurement mode and state vector composition, by including the respective elements in $\mathbf{K}$, $\mathbf{S}_y$ and $\mathbf{S}_a$. Table 6 shows the average DOFS obtained for each retrieval parameter. We recall that concentration profiles $c_s$ are retrieved at 25 altitudes and that $\omega_{\mathrm{surf}}$, $n_1$ and $n_2$ are retrieved at six wavelengths. Here, the individual DOFS are summed up, therefore yielding values $> 1$. The number of summed state vector elements, and hence the maximum possible number of DOFS, is indicated in the final row of this table. Figure 3 visualises these results for selected measurement modes.





**Table 5.** Overview of measurement mode labels.

| Flag | Description |
|---|---|
| | Spectral band of the observations: |
| UV | "UV" indicates that only observations at 343 and 360 nm go into the retrieval. |
| Vis | "Vis" uses observations at 415, 460, 477 and 532 nm. |
| Multi | "Multi" uses observations at all wavelengths. |
| S | Indicates a simultaneous retrieval of all species in a common model atmosphere. In contrast to the other flags, this is a retrieval setting and therefore has no impact on the composition of the measurement vector. If "S" is not contained in the mode description, species are retrieved separately: first, aerosol concentrations and properties as well as surface albedo are inferred from $O_4$ dSCDs and dSOTs, then each trace gas profile is retrieved in the resulting atmosphere with all other parameters being fixed. |
| P | Indicates the incorporation of polarimetric information. No "P" in the mode code indicates that only observations with no polariser ($\delta = -$) are considered. Otherwise, DSCDs at all PAs except $\delta = -$ are incorporated, each evaluated against the reference SCD at $\alpha = 90°$ and $\delta = 0°$. Further, to capture broadband polarisation features, dSOTs at $\delta \in \{60, 120°\}$ are incorporated, each evaluated against the SOT in the same viewing direction and $\delta = 0°$ (for further explanation see Section 5.2). |
| I | Indicates that spectral broadband features between different viewing directions serve as additional sources of information. In this case, dSOTs at $\delta = 0°$ are incorporated, each evaluated against the SOT at $\alpha = 90°$ and $\delta = 0°$ (for further explanation, see Section 5.2). |
| A | Indicates the incorporation of an almucantar scan, performed under the same solar geometries and atmospheric conditions as the elevation scan. The effects of 'S' and 'P' described above also apply for the almucantar dSCDs and dSOTs. |

The effect of including polarimetric information (flag P) for different cases can be inferred by comparing UV-S and UV-S-P,
Vis-S and Vis-S-P, Multi-S and Multi-S-P and so on. A significant increase in information is observed for all these modes when adding polarimetric information, particularly on aerosol properties, aerosol profiles and surface albedo. For the Multi-S-P mode, the increase is about 1.2 DOFS (60 %) for the aerosol profile, 0.5 DOFS (20 %) for trace gas profiles, 1.3 DOFS (190 %) in the surface albedo and 4.7 DOFS (80 %) for aerosol properties.

We further calculated the DOFS for a reduced state vector $\mathbf{x}$ comprising just the concentration profiles, with the underlying
aerosol parameters and surface albedos being set to their "true values" as listed in Table 4. DOFS results for this case are presented in Table S1 in the supplement. We see that increases in DOFS for the concentration profiles are significantly smaller (by about 50 % compared to the case with all parameters being retrieved), indicating that a large fraction of the increase in information on the profiles when retrieving the full state vector is an indirect effect of the improved knowledge on aerosol properties and surface albedo.

In addition to the information shown in the DOFS tables, it is worth discussing the individual effects of the incorporation of polarimetric dSCDs and polarimetric dSOTs, respectively. Exact DOFS values for these cases are given in Table S2 in the supplement. While polarimetric dSCDs predominantly increase information content on vertical profiles, polarimetric dSOTs



**Table 6.** DOFS achieved with different measurement modes (table rows) for individual state vector elements (table columns) averaged over the simulated atmospheric scenarios, namely all combinations of aerosol profiles, trace gas profiles, SZAs and RAAs. The final row lists the total parameter numbers, and hence the maximum possible DOFS values.

| Measurement mode | | | | | Total | Profiles | | | Surface | Aerosol properties | | | | | | | | |
|---|---|---|---|---|---|---|---|---|---|---|---|---|---|---|---|---|---|---|
| Band | S | P | A | I | | $c_{aer}$ | $c_{HCHO}$ | $c_{NO_2}$ | $\omega_{surf}$ | $r_1$ | $r_2$ | $\sigma_1$ | $\sigma_2$ | $\Re n_1$ | $\Re n_2$ | $\Im n_1$ | $\Im n_2$ | $f$ |
| UV | ✗ | ✗ | ✗ | ✗ | 8.6 | 1.21 | 1.93 | 2.97 | 0.17 | 0.65 | 0.04 | 0.21 | 0.09 | 0.81 | 0.0 | 0.33 | 0.02 | 0.21 |
| UV | ✓ | ✗ | ✗ | ✗ | 8.9 | 1.51 | 1.76 | 2.59 | 0.24 | 0.68 | 0.05 | 0.24 | 0.11 | 1.01 | 0.0 | 0.43 | 0.03 | 0.24 |
| UV | ✓ | ✓ | ✗ | ✗ | 12.5 | 2.5 | 2.14 | 3.11 | 0.72 | 0.77 | 0.08 | 0.43 | 0.19 | 1.23 | 0.01 | 0.77 | 0.06 | 0.42 |
| Vis | ✗ | ✗ | ✗ | ✗ | 7.2 | 1.3 | 0.0 | 2.89 | 0.18 | 0.65 | 0.05 | 0.36 | 0.09 | 0.82 | 0.0 | 0.47 | 0.06 | 0.31 |
| Vis | ✓ | ✗ | ✗ | ✗ | 7.5 | 1.52 | 0.0 | 2.31 | 0.26 | 0.66 | 0.05 | 0.37 | 0.1 | 1.26 | 0.0 | 0.57 | 0.08 | 0.33 |
| Vis | ✓ | ✓ | ✗ | ✗ | 11.1 | 2.65 | 0.0 | 2.79 | 0.83 | 0.75 | 0.08 | 0.57 | 0.15 | 1.57 | 0.04 | 0.86 | 0.22 | 0.59 |
| Multi | ✓ | ✗ | ✗ | ✗ | 13.5 | 1.99 | 1.77 | 2.89 | 0.67 | 0.73 | 0.07 | 0.51 | 0.16 | 2.77 | 0.01 | 1.32 | 0.15 | 0.44 |
| Multi | ✗ | ✓ | ✗ | ✗ | 20.8 | 2.89 | 2.34 | 3.77 | 1.62 | 0.85 | 0.1 | 0.72 | 0.23 | 4.95 | 0.13 | 2.12 | 0.42 | 0.63 |
| Multi | ✓ | ✓ | ✗ | ✗ | 21.9 | 3.24 | 2.21 | 3.43 | 1.95 | 0.87 | 0.11 | 0.75 | 0.26 | 5.22 | 0.15 | 2.54 | 0.5 | 0.65 |
| Multi | ✓ | ✗ | ✓ | ✗ | 18.2 | 2.66 | 1.99 | 3.2 | 1.57 | 0.86 | 0.24 | 0.82 | 0.69 | 2.96 | 0.05 | 1.95 | 0.29 | 0.92 |
| Multi | ✓ | ✓ | ✓ | ✗ | 27.4 | 3.91 | 2.5 | 3.77 | 2.95 | 0.97 | 0.31 | 0.95 | 0.78 | 5.62 | 0.5 | 3.21 | 0.99 | 0.96 |
| Multi | ✓ | ✗ | ✗ | ✓ | 20.8 | 2.46 | 1.83 | 2.99 | 1.62 | 0.81 | 0.11 | 0.69 | 0.24 | 5.15 | 0.04 | 3.86 | 0.45 | 0.55 |
| Multi | ✓ | ✓ | ✓ | ✓ | 32.6 | 4.12 | 2.55 | 3.83 | 3.93 | 0.99 | 0.71 | 0.98 | 0.93 | 5.93 | 0.97 | 5.12 | 1.52 | 0.98 |
| | | | | | 110 | 25 | 25 | 25 | 6 | 1 | 1 | 1 | 1 | 6 | 6 | 6 | 6 | 1 |

increase information mainly on aerosol properties. However, polarimetric dSCDs and dSOTs also carry significant amounts of equivalent information: starting from the Multi-S mode, the incorporation of polarimetric dSCDs increases the total DOFS from 13.5 to 17.8, whereas including only polarimetric dSOTs yields 18.8 DOFS. Including both yields the Multi-S-P mode with 21.9 DOFS.

Figure 4 shows the relative increase in DOFS ($\Delta$DOFS) between the Multi-S and the Multi-S-P measurement mode, grouped according to different aerosol scenarios. The effective light paths are generally determined by the atmospheric aerosol content. Dependence on the trace gas profiles is not shown here, but was found to have a minor effect, at least for optically thin trace gases. $\Delta$DOFS is particularly large for the Aer2 scenario, which features the largest aerosol VCD (exponential profile with $2 \times 10^8\,\mu m^2\,cm^{-2}$). A particularly small $\Delta$DOFS is observed for the Aer0 (no aerosol) scenario. Of course, aerosol properties cannot be retrieved in this case, but also the information increase for the aerosol profile is significantly lower than for other scenarios. For aerosol scenarios with the same VCD, the variation in $\Delta$DOFS is rather small, indicating that the benefit from the inclusion of polarimetric observations is mostly independent of the profile shape. However, there are indications that having a distribution of aerosols over a large altitude range is advantageous for the retrieval of aerosol properties.

Figure 5 shows DOFS for vertical profiles, obtained for the concentrations of different species in each retrieval layer. The DOFS profiles show a large relative increase in information on aerosol concentrations at higher altitudes between 1 and 4 km.




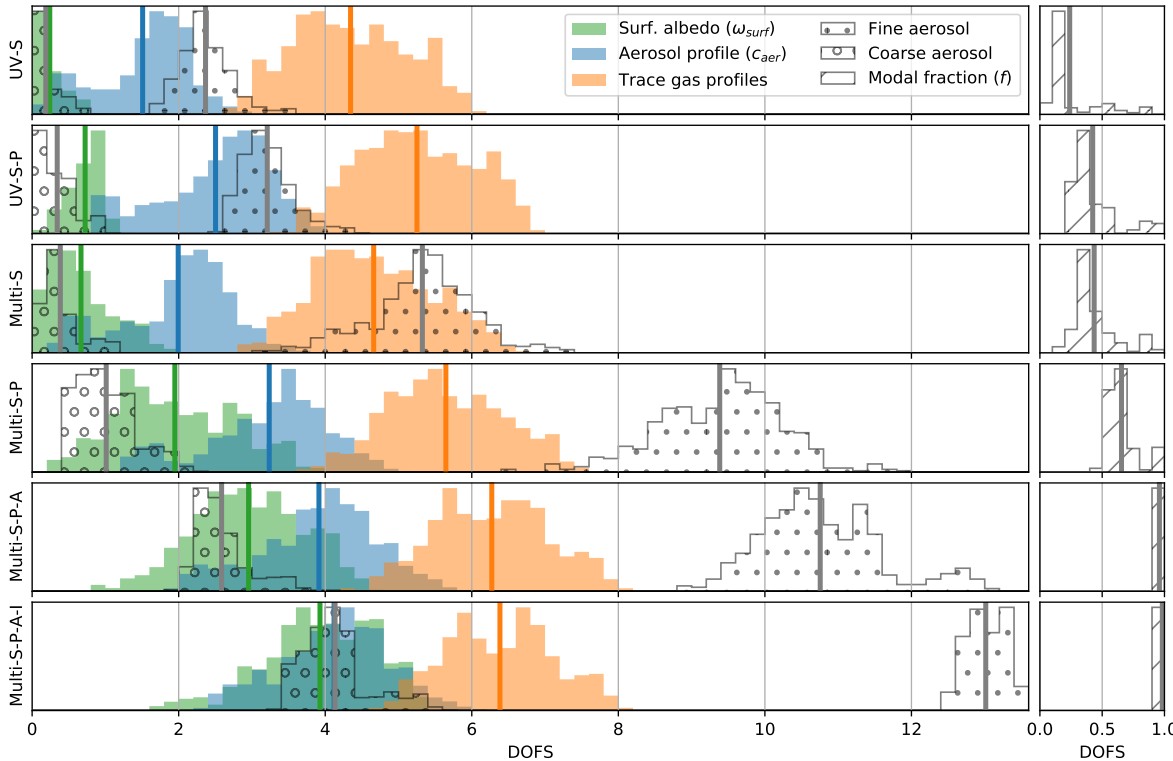

**Figure 3.** Visualisation of the DOFS as shown in Table 6. Shown are histograms of the DOFS obtained for different parameter subgroups (see legend). Each occurence corresponds to one atmospheric scenario. Trace gas profile DOFS for $NO_2$ and HCHO have been summed up. Subplot rows show different measurement modes according to the labels on the left axes. The histograms are peak-normalised, so that the vertical axes indicate the number of occurences in arbitrary units. For extra clarity, histograms for the modal fraction $f$ are shown in separate panels on the far right of the figure. The aerosol scenario Aer0 (no aerosol) was excluded here.

However, the absolute values are still rather small ($< 0.2$), so that aerosol abundances at these altitudes are still barely retrievable even with the inclusion of polarimetric MAX-DOAS observations.

The effect of the simultaneous retrieval of aerosol and trace gases (flag S) can be inferred by comparing the measurement modes UV and UV-S as well as Vis and Vis-S. As expected, the information content in aerosol profiles and properties is slightly enhanced (by about 0.3 and 0.4 DOFS, respectively), since both the $O_4$ dSCDs and the trace gas dSCDs are sensitive to aerosols (aerosols affect slant light paths). On the other hand, information on trace gas concentrations is reduced ($\approx -0.4$ DOFS), since the retrieval now encompasses cross-sensitivity to aerosol parameters with concomitant propagation of retrieval errors. Both

effects have been absent in earlier MAX-DOAS retrieval algorithms, the latter effect causing an underestimation of the trace gas profile error. The gain in information slightly prevails, resulting into an increase of about 0.3 DOFS in the total information content.





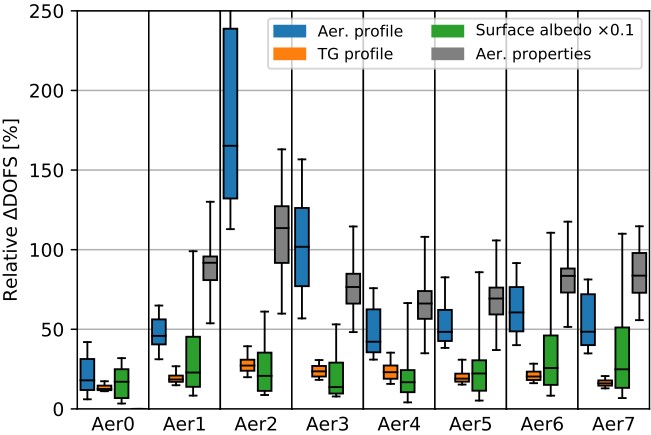

**Figure 4.** Box-whisker plots of relative DOFS increases resulting from inclusion of polarimetric information, grouped by different aerosol profile scenarios (different subplots) and parameter sub-groups (box colors). $\Delta$DOFS is defined here as the difference in DOFS values obtained with the polarimetric (Multi-S-P) and non-polarimetric (Multi-S) measurement modes.

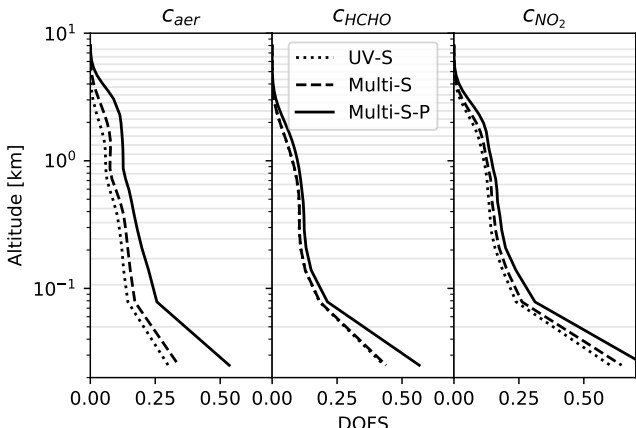

**Figure 5.** Average DOFS vertical profiles for aerosol (left), HCHO (center) and $NO_2$ (right) concentrations. Different measurement modes were simulated as indicated in the legend. Horizontal grey lines indicate model layer boundaries.

The information content in the measurements can further be enhanced by including almucantar scans (flag A) and spectral broad-band features (flag I). Our investigations show that, compared to conventional non-polarimetric monochromatic retrievals (even when summing the DOFS obtained for the UV and Vis mode), the total information content of MAX-DOAS observations can be more than doubled by including multispectral, polarimetric dSOT and dSCD observations under optimised viewing geometries. In general, information content on coarse mode aerosol properties remains low (less than 5 of 14 possible DOFS)





for all measurement modes, even for the Multi-S-P-A-I mode. The impact of the aerosol size distribution on the DOFS is discussed in more detail below.

We further investigated the manner in which the DOFS depend on aerosol VCD. Simulations at $\theta = 60°$, $\phi = 90°$ for different aerosol VCDs were performed, assuming an exponential $c_{\text{aer}}$ profile with a scale height of $1\,\text{km}$. The TG1 scenario was chosen for the trace gas profiles. DOFS values for the three measurement modes and three parameter subgroups are illustrated in Figure 6. For small AOTs ($< 0.2$) the DOFS values for profiles remain constant. For higher values, the horizontal visibility range in the atmosphere and thus the sensitivity decreases. The DOFS for the surface albedo is generally low for

non-polarimetric measurements. For multispectral measurements, highest values (DOFS of $\approx 5$ for albedos at six different wavelengths) are obtained for low aerosol loads. For aerosol properties, there are two limiting factors: with decreasing aerosol abundance, of course the sensitivity towards aerosol properties decreases. On the other hand, DOFS also decrease with high AOTs, again due to the reduced horizontal visibility range in the atmosphere. Maximum DOFS are achieved at AOTs around unity.

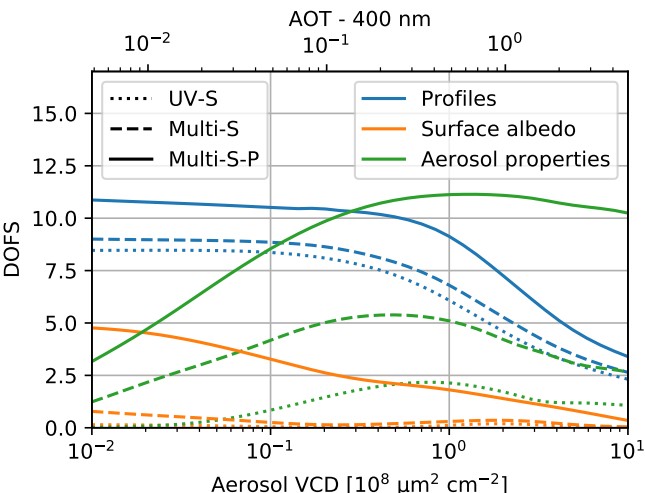

**Figure 6.** Dependence of DOFS on aerosol VCD for different measurement modes. For the detailed settings, see main text.

Figure 7 shows the dependence of DOFS on the aerosol size for monomodal Mie aerosol with properties as defined for the fine mode in Table 4, but with different mode radii $r_1$. We assume profile scenarios Aer1-TG1, $\theta = 60°$ and $\phi = 90°$. Note that the aerosol surface area is kept constant, whereas the AOT varies with $r_1$, due to changes in aerosol scattering efficiency. For the UV-S and the Multi-S-P mode, there is a limited number of particle size ranges having significant sensitivity to all aerosol parameters. This is expected to some degree; the dependence of aerosol extinction efficiency on particle size parameter

exhibits an abrupt rise at size parameters close to unity (the transition between Rayleigh and Mie scattering). The shape of this rise strongly depends on the aerosol microphysical properties, and thus observations at corresponding wavelengths are expected to yield most information. Consequently, for the Multi-S-P mode (which also includes the larger Vis wavelengths),



the upper limit of the $r_1$-range for which good information can be obtained is higher than that obtained with UV-S. By adding
an almucantar scan (A), the range cannot be further extended (not shown in the figure). In contrast, including (P), (A) and
(I) simultaneously yields DOFS close to unity for all aerosol parameters and sizes up to $r_1 = 2\,\mu m$. Recall, however, that in
Figure 7 we consider monomodal aerosol. The general lack of information on coarse aerosol indicated in Figure 3 suggests that
in the bi-modal case the modes are barely distinguishable. Sensitivity of the size range could be enhanced by further extending
the spectral range of observations towards higher wavelengths. Interestingly, all measurement modes shown yield very good
information on $r_1$ for small radii, while the information on the modal width increases with $r_1$.

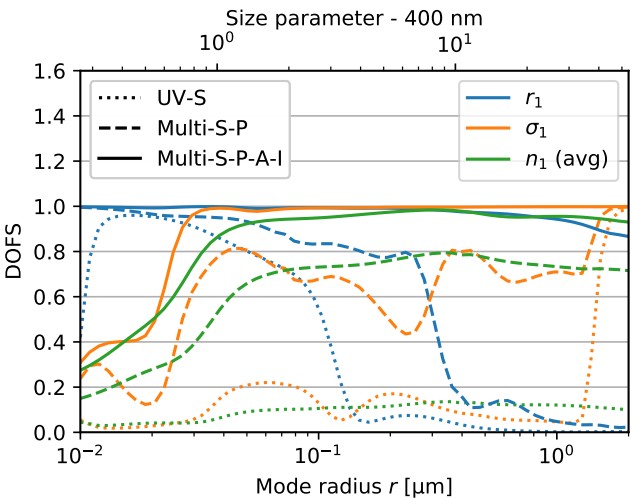

**Figure 7.** Dependence of DOFS on the aerosol size for different measurement modes. $n_1$ are the average DOFS per refractive index parameter
(comprising real and imaginary parts at multiple wavelengths). The size parameter on the top axis was calculated from the mode effective
radius. For details on the settings, see main text.

## 8    Considerations on viewing geometries


The information content of polarimetric MAX-DOAS observations is expected to depend on the set of viewing geometries,
at which measurements are provided. Figure 8 shows the dependence of DOFS (for the Multi-S-P mode) on the SZA and
the elevation scan RAA. Interestingly, for both aerosol and trace gas profiles the dependence is rather weak, whereas for the
surface albedo and aerosol properties there is a stronger dependence with peak-to-peak changes of approximately 2 DOFS.
The total information content is lowest for elevation scans close to the sun, where the single scattering angles realised over
the elevation scan are smaller than those at larger RAA values. This suggests that polarimetric information is maximised at
viewing directions with single scattering angles close to $90°$ (where the largest DOLP values are expected). This is certainly





the case for all EAs for instance, if SZAs are large and elevation scan RAAs are close to $90°$. In fact, for SZA $\theta = 80°$, (local) maxima in DOFS can be observed around RAAs $\phi \approx 90°$ for all parameter subgroups.

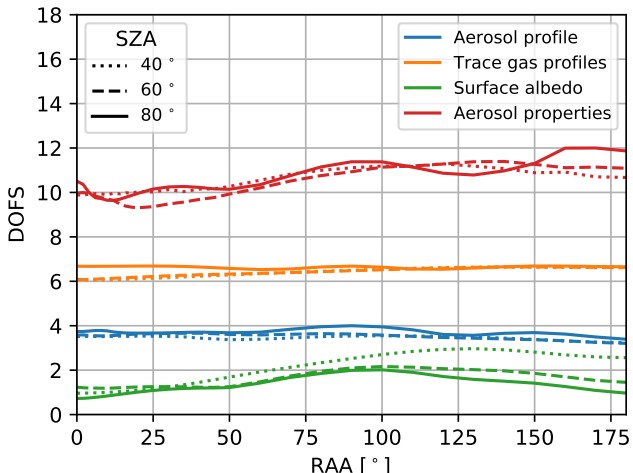

**Figure 8.** Dependence of the DOFS for the Multi-S-P mode on the viewing geometry, namely different SZAs (line styles) and elevation scan RAAs (x-axis). Simulations were performed for the profile scenario Aer1-TG1.

The question arises, whether elevation scans performed at a fixed RAA are the ideal measurement geometry. To investigate this further, we tested information content with sets of measurements made with a "tilted elevation scan" geometrical configuration; for this, we assume the same set of EAs as before, but calculate RAAs for each EA according to

$$
\phi_{90} = \begin{cases} \arccos\left(-\frac{\tan\alpha}{\tan\theta}\right), & \text{if } \alpha < \theta \\ 180°, & \text{if } \alpha \geq \theta \end{cases} \tag{25}
$$

Here, $\phi_{90}$ represents the RAA at which the single scattering angle is closest to $90°$ for given SZA $\theta$ and EA $\alpha$. The resulting
geometry for $\theta = 40°$ is illustrated in Supplement S5. The total DOFS obtained for such a tilted elevation scan, assuming Aer1-TG1 profiles and measurement mode Multi-S-P yields $23.8$. This is slightly smaller than the total DOFS of $24.4$, achieved with a conventional vertical elevation scan at $\phi = 140°$ for the same atmospheric conditions (see Figure 8). This suggests that the tilted elevation scan configuration has no advantage over conventional elevation scans performed at fixed RAA.

So far, we have assumed that measurements are performed at three fixed PAs $\delta \in \{0°, 60°, 120°\}$. However, the DOFS
can be increased if measurements are performed a only two optimised PAs. For many viewing directions and conditions, the orientation of the skylight polarisation $\chi$ can be reasonably predicted, since it is approximately perpendicular to the Sun's incident angle $\theta'$ into the instrument FOV. It can be shown that

$$
\tan\theta' = \frac{\sin\alpha\cos\phi\sin\theta - \cos\alpha\cos\theta}{\sin\phi\sin\theta}, \tag{26}
$$





where $\theta'$ is given with respect to the horizon, increasing clockwise when looking towards the instrument. Consequently, we
find

$$\chi \approx \theta' + 90°. \tag{27}$$

Thus, two spectra recorded with the polariser's transmitting axis parallel and perpendicular to the predicted value of $\chi$ are then sufficient to constrain firmly the state of polarisation of skylight, and these two spectra should yield nearly the same information as that achieved with three PAs. In fact, simulations for $\theta = 60°$, $\phi = 90°$, Mie aerosol and Aer1-TG1 profiles
yield almost the same value of total DOFS for both approaches: with the two optimised PAs one achieves 22.7 DOFS, with three PAs $60°$ apart, the figure is 23.0 DOFS. The approach with two angles improves the temporal resolution by a factor $2/3$. Alternatively, the exposure times per spectrum could be enhanced by a factor $3/2$, which yields a corresponding gain in light, thus resulting in reduced dSCD uncertainties by a factor $\sqrt{2/3}$ (assuming that DOAS analysis results are photon shot-noise limited). Considering this, one obtains an increase in total DOFS up to 23.5.

To further improve temporal resolution of MAX-DOAS measurements, it is to determine to what extent the number of viewing directions might be reduced without significant loss of information. To test this idea, we used measurement mode Multi-S-P-A, Mie aerosol, $\theta = 60°$, $\phi = 90°$, Aer1-TG1 and calculated the total DOFS for two cases: (1) with the full elevation and almucantar scans according to Table 3 and (2) with reduced elevation (EA $\in \{1, 2, 5, 10, 30, 90°\}$) and solar almucantar scans ($\phi \in \{2, 2.5, 3., 3.5, 4, 5, 7, 10, 15°\}$). The two cases yield total DOFS of 28.2 and 27.6, respectively. This is a surprisingly
small decrease in information, considering the strong reduction in the number of viewing directions (from 38 to 15).

From the results presented in this section we conclude that there is a high potential for the optimisation of measurement geometries. Looking ahead, optimal geometries for different numbers of viewing directions might be investigated by performing more comprehensive studies in this direction.

## 9  Retrieval results

In this Section, synthetic observations are used as input $\hat{\mathbf{y}}$ for RAPSODI to perform actual inversions. This was done for a reduced dataset of observations corresponding to seven combinations of aerosol and trace gas profiles (Aer1-TG6, Aer2-TG5, Aer3-TG4, Aer4-TG1, Aer5-TG2, Aer6-TG7 and Aer7-TG3), for $\theta = 60°$, $\phi = 90°$. As before, different measurement modes are realised by providing different sets of observations $\hat{\mathbf{y}}$ to RAPSODI. Each inversion was performed eleven times: ten times (to obtain some statistics) with a random noise component added to $\hat{\mathbf{y}}$ with standard deviations according to the assumed
uncertainties in Table 1 and once more with $\hat{\mathbf{y}}$ having no noise (exact simulated observations). Figures 9 and 10 show the retrieval results for two selected Aer-TG-scenarios, comparing the ground truth, a priori and retrieved state vector elements. Results for the remaining Aer-TG-scenarios are shown in Figures S6 to S10 in the supplement. Figure S11 in the supplement illustrates the quality of the retrieval convergence (comparison of the input observations with the modelled observations for the retrieved atmospheric states). Figure 11 shows a statistical representation of all retrieval results.

Generally, all retrievals converged well (see Figure S11): deviations between $\hat{\mathbf{y}}$ and $\mathbf{F}(\hat{\mathbf{x}})$ are mostly within the measurement uncertainties that define the magnitude of the synthetic random noise added to $\hat{\mathbf{y}}$. Interestingly, these deviations are largest for





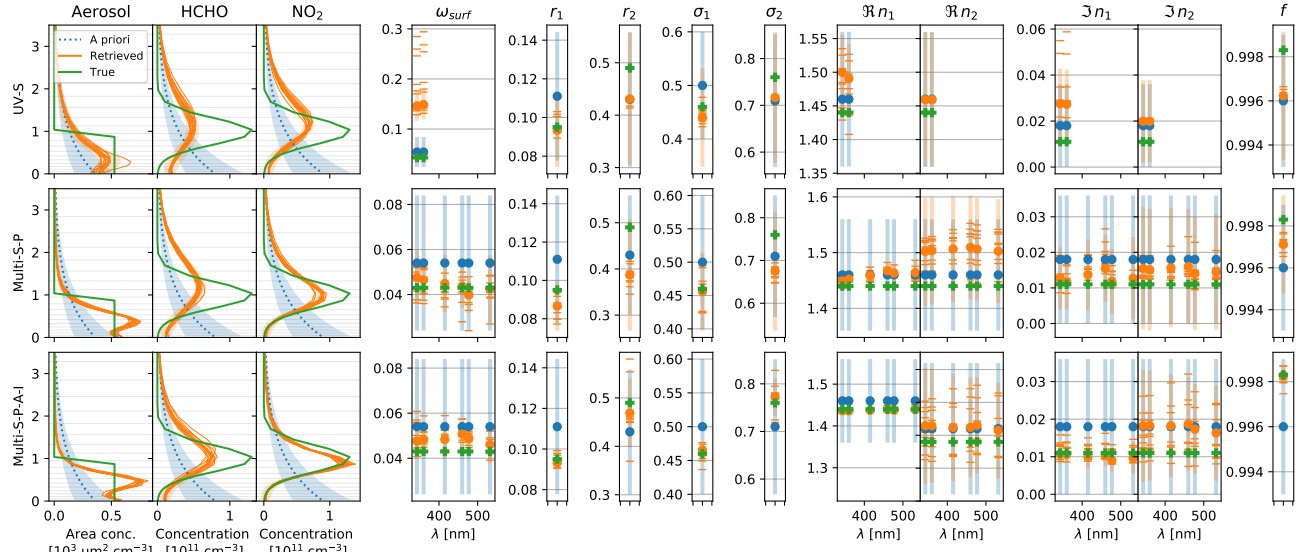

**Figure 9.** Retrieval results for the Aer6-TG7 scenario. Different subplot rows show different measurement modes (see captions on the left). The first three subplot columns show profiles with altitude in km on the y-axis. Remaining columns show values (y-axis) of other state vector elements (see captions at the top), occasionally for different wavelengths (x-axis). Blue lines and symbols show a priori values, green lines and crosses show true values. Thick orange lines and circles indicate results for a noiseless retrieval while thin lines and dashes indicate the results of the ten retrievals with random noise added to the observations. Transparent areas and bars indicate uncertainties.

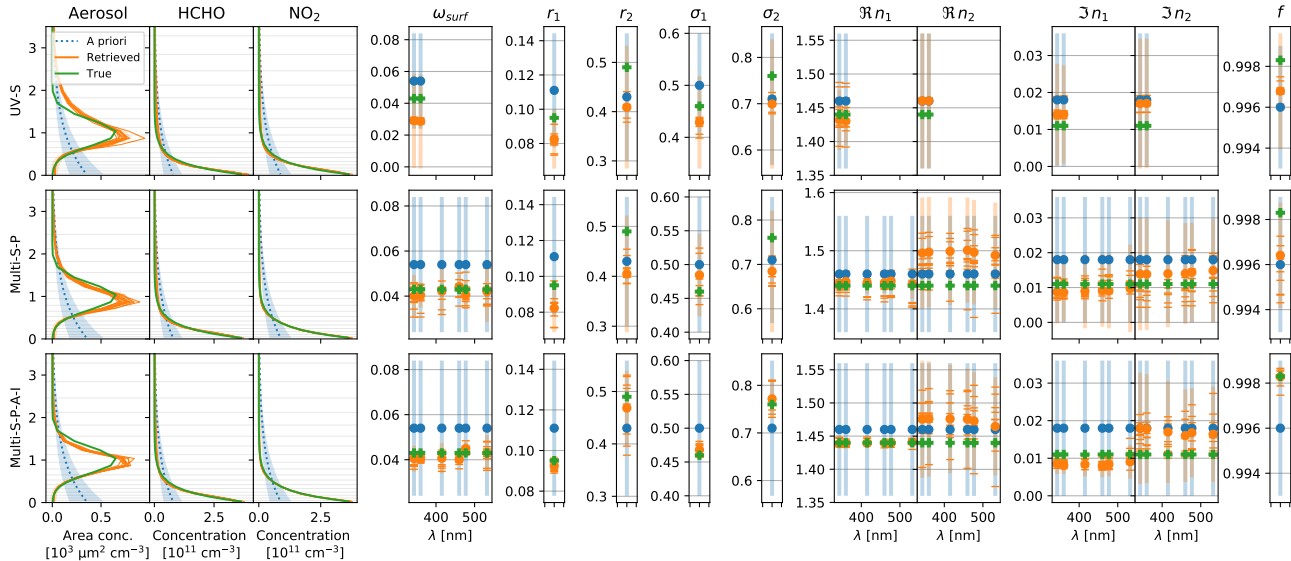

**Figure 10.** Retrieval results for the Aer7-TG3 scenario. Description of Figure 9 applies.





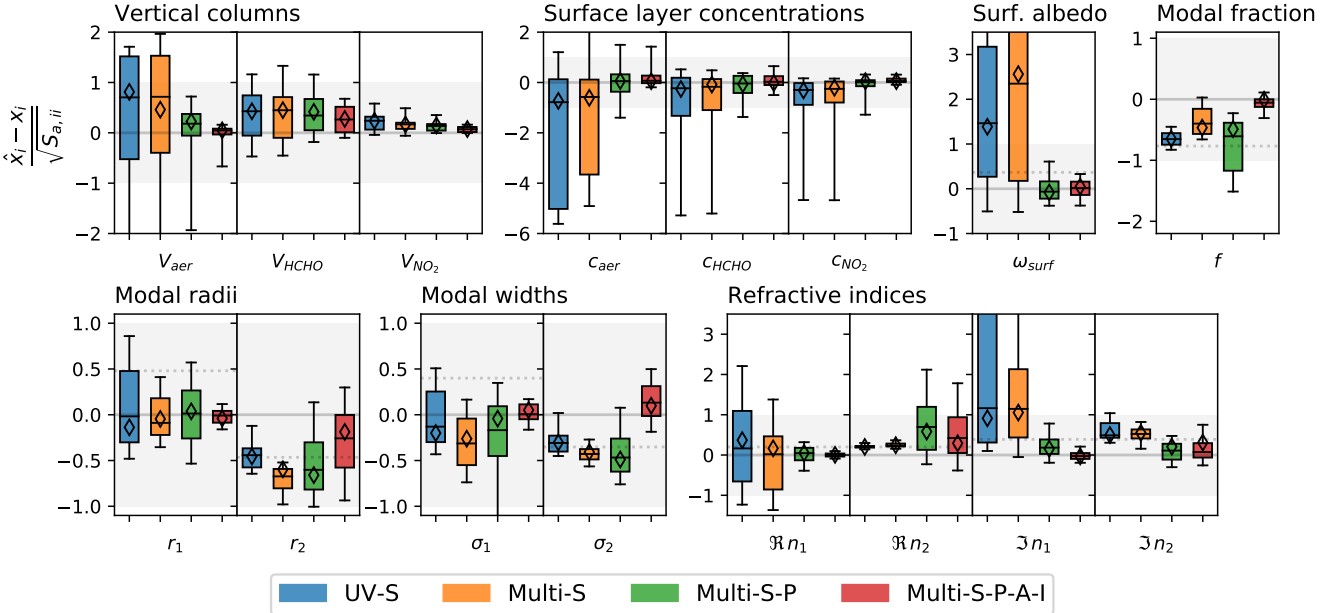

**Figure 11.** Statistical representation of all retrieval results, grouped by parameter (different subplots) and different measurement modes (box colours according to legend). The y-axis represents the difference to the true value, normalised by the a priori uncertainty. The latter is indicated by the grey shaded areas. Where possible, dotted grey lines indicate the a priori value. Deviations $\ll 1$ with little bias towards the a priori indicate a successful retrieval. Boxes span the 25th to 75th, whiskers the 5th to 95th percentile. Dashes indicate the median, diamonds indicate the median of the retrievals from noiseless observations.

the UV-S case even though, compared to the other modes, the retrieval fits fewer measurements by optimising the same set of state vector elements. The same is true when comparing Multi-S-P and Multi-S-P-A-I. This indicates that fewer observations increase the risk for the inversion to end up in a local minimum of the OEM cost-function. This is supported by the retrieval

results in Figure 11: particularly for the UV-S mode, the retrieval sometimes diverges (e.g. for $\omega_{surf}$, $\omega_{hg}$ and $\Im n_1$) instead of converging to the a priori values (e.g. in the case of $n_2$ or $f$). This phenomenon might be related to the non-linearity of the inverse problem. Nevertheless, the results mostly behave as one would expect from the information content analysis in Section 7. The example retrieval results shown in Figures 9 and 10 demonstrate that the major features of the profiles are well captured by the retrieval, particularly with the Multi-S-P and the Multi-S-P-A-I modes. The incorporation of polarisation

(comparison between Multi-S and Multi-S-P mode) significantly improves the results for aerosol columns and surface concentrations. Further it strongly stabilises the retrieval with respect to surface albedo and fine mode refractive indices. For some microphysical properties (e.g. $r_1$, $\sigma_1$, $f$) the results degrade. This can partly be attributed to stronger biases towards the a priori for the Multi-S mode, leading to more stable results. Nevertheless, the overall RMSD (root-mean-square deviation) between retrieved and true aerosol property values is reduced by about $40\,\%$, this being mainly due to the strong improvement in the

refractive index retrievals (note that they are retrieved at multiple wavelengths). In conclusion, the retrieval of microphysical



properties is feasible, but useful results are mostly achieved only for fine mode properties and only when the Multi-S-P-A-I mode is in force.

## 10  The effect of increased noise for polarimetric observations

So far we have assumed equal uncertainties for non-polarimetric and polarimetric observations. This can only be achieved for polarimetric observations if the instrument has a higher light throughput or if measurement times are longer than those required for measurements without polarisation sensitivity. However, if the same type of instrument applies and a similar time resolution is desired, two aspects need to be considered: 1) for the type of instrument assumed for this work, on average half of the light entering the telescope will be rejected by the linear polariser and 2) for non-polarimetric dSCDs, a single spectrum has to be recorded at each viewing direction, whereas for polarimetric observations as assumed above, three spectra need to be recorded in the same time period (neglecting any time losses due to repositioning of the polariser). Hence, regarding individual polarimetric observations, only a sixth of the light will be available compared to the non-polarimetric equivalents (the effective loss of light is only a factor of two, and there are three times as many polarimetric measurements ). With the precision of MAX-DOAS dSCDs typically limited by photon shot noise, the uncertainty of individual polarimetric dSCDs is therefore expected to increase by a factor $\sqrt{6}$. The situation is different for dSOTs; here the initial uncertainty of $2\%$ is assumed to be dominated by systematic effects (e.g. instrumental misalignment) and is much larger than typical photon shot noise uncertainty in DOAS applications ($\approx 10^{-4}$). Hence, for dSOTs the loss of light is expected to have negligible impact the measurement accuracy.

From a practical stand-point, there are several considerations to bear in mind:

1. As discussed in Section 8, the polariser positions $\delta$ could be optimised during the measurement in order to enhance the information content per spectrum, as opposed to the approach with three fixed polariser positions taken here.

2. MAX-DOAS dSCDs are prone to systematic errors, for instance due to uncertainties in the literature cross-sections, instrumental effects or simplifying assumptions in the DOAS spectral analysis. Further, deviations between measured and modelled dSCDs that are much larger than the actual measurement accuracy might occur because of horizontal and temporal variability in the atmosphere.

3. Alternative instrumental setups detecting two polarisation directions simultaneously are conceivable.

To this end, we have repeated the above investigations assuming an increased uncertainty (factor $\sqrt{6}$) for all polarimetric dSCDs. The results are shown in Section S5 in the supplement. However, for the reasons given above, the truth might lie somewhere between the idealised siutation and the case with increased uncertainty. As expected, the information gain obtained from polarimetric observations is generally lower under enhanced noise conditions, in general this varies by different amounts for different parameter subgroups. The gain in information on aerosol properties is not much affected ($\Delta$DOFS between Multi-S and Multi-S-P mode decreases by about $15\%$), since a large part of the information here is inferred from the polarimetric dSOTs, rather than from the dSCDs. In contrast, the information gain for the vertical profiles is significantly reduced. For aerosol profiles, the increase in DOFS between the Multi-S and the Multi-S-P measurement mode is $\Delta$DOFS $= 0.6$, instead of





$\Delta$DOFS $= 1.2$ (as reported in Section 7). In particular, the use of polarimetric observations even reduces the information on trace gas profiles for both HCHO ($\Delta$DOFS $\approx -0.12$) and NO$_2$ ($\Delta$DOFS $\approx -0.08$), indicating that the disadvantage caused

by reduction of light due to the presence of the polariser overrides any advantages to be gained through the use of polarimetry. Figure S13 shows a degradation of the retrieval accuracy for the polarisation incorporating measurement modes; nevertheless, the main conclusions drawn in Section 9 remain qualitatively valid.

## 11   Effect of spatio-temporal variability in atmospheric composition

The forward models in MAX-DOAS retrievals assume horizontally homogeneous atmospheres over typical MAX-DOAS hor-
izontal sensitivity ranges of several kilometres. Furthermore, a single atmospheric state is retrieved from observations acquired over time periods of several minutes. Clearly, spatio-temporal variability can in principle cause deviations between modelled and measured observations that are much larger than actual measurement uncertainties. Ideally, the retrieved atmospheric state in such cases corresponds to a kind of spatio-temporal average. However, large variability is expected to induce significant biases in the results or even prevent the inversion to achieve reasonable convergence. In order to investigate the impact of
spatio-temporal variability on the retrieval results, we simulated inhomogeneity by adding random noises of different relative magnitude (0, 2, 4, 6, 8 and $10\%$) to the observations $\hat{\mathbf{y}}$ (in addition to the noise that is expected from the measurement uncertainties) before performing the same retrievals as those discussed in Section 9, but this time limited to the Aer1-TG1 case. This "inhomogeneity noise" was assumed identical for all observations in the same viewing direction. For each inhomogeneity magnitude level, ten retrievals were performed, each time recalculating the random noise pattern. To put the noise magnitudes
into perspective: Frieß et al. (2019) for instance assume an inhomogeneity noise of $5\%$, this figure motivated by observations performed during the CINDI-2 campaign (Kreher et al., 2019). Figure S14 in the supplement shows a statistical representation of the results for the Multi-S-P and Multi-S-P-A-I measurement modes. In the most problematic cases - for instance coarse-mode aerosol properties and surface albedo - the results already exceed the a priori uncertainty (grey shaded area) at noise levels of a few percent. In contrast, the effect on trace gas VCDs and surface concentrations is small. Also the retrieval of
fine-mode aerosol properties still yields useful results (deviations from the truth are smaller than the a priori uncertainty) in the presence of inhomogeneities, at least for the Multi-S-P-A-I mode.

## 12   Conclusions

In this work, we have developed and tested a novel retrieval algorithm (RAPSODI), capable of processing polarimetric MAX-DOAS observations; the algorithm utilises the corresponding information to retrieve vertical distributions of aerosol and trace
gases as well as aerosol properties. Furthermore, in contrast to earlier MAX-DOAS algorithms, RAPSODI retrieves all species of interest simultaneously in a shared model atmosphere, enabling us to infer aerosol microphysical properties through the use of a Mie scattering model.





Based on the use of synthetic measurement data, the algorithm was used to assess the potential of polarimetry in ground-based MAX-DOAS applications. Our results suggest that polarimetric information significantly increases the total information content of MAX-DOAS observations (about $50\%$ increase in total DOFS), in particular for aerosol related quantities. Table 7 provides a succinct summary: it shows the relative improvement in information content (in terms of DOFS) and retrieval accuracy (in terms of RMSD between the retrieval results and the true values) for crucial parameters when going from non-polarimetric (Multi-S mode) to polarimetric (Multi-S-P) retrievals from elevation-scan data. All atmospheric parameters are retrieved (comprising aerosol profiles, trace gas profiles and aerosol microphysical properties, surface albedo and aerosol mode fraction). DOFS values represent averages over all atmospheric scenarios and viewing geometries as described and applied in Section 6 and 9. In the table we distinguish between an idealised case, which assumes the same measurement error for non-polarimetric and polarimetric observations (compare Section 7) and an increased-noise case, in which an increased noise (factor $\sqrt{6}$) is assumed for the polarimetric dSCD observations (compare Section 10). The increase in total DOFS (sum over DOFS for all parameters) ranges between $40\%$ (increased-noise) and $60\%$ (idealised case). The increase in information is largest for aerosol-related quantities, whereas for trace gas profiles, the DOFS enhancement is generally small and even degrades for the increased-noise case. Similar patterns emerged in the retrieval accuracy: the RMSD decrease is largest for aerosol related quantities. A notable exception lies with concentrations in the surface layer, where the accuracy improves for all species (including trace gases) by about $70\%$ and $40\%$ in the idealised and the increased-noise cases, respectively. It seems that accurate knowledge of aerosol loading is much more important for the retrieval of surface trace gas concentrations than for the retrieval of trace gas VCDs.

In the Multi-S-P mode and assuming the idealised noise case, fine- and coarse-mode parameters can be retrieved to accuracies of $30\%$ and $70\%$ of the a priori uncertainty, respectively. However, some coarse-mode parameters ($r_2$, $\Im n$ and also $f$) are still strongly biased towards the a priori, owing to limited information inherently available in the measurements. When a solar almucantar viewing configuration and broad-band spectral information are also incorporated (Multi-S-P-A-I mode), the information can be further increased by about $50\%$ (relative increase in total DOFS compared to the Multi-S-P mode), with the largest increase for aerosol microphysical properties ($70\%$). Fine -and coarse-mode parameters can then be retrieved to accuracies of $10\%$ and $30\%$ of the a priori uncertainty, and a priori biases for coarse-mode parameters are strongly reduced.

Things change again when potential inhomogeneities in the atmosphere are taken into account (simulated by adding an additional noise of up to $10\%$ to the observations). In particular, the retrieval of surface albedo and coarse-mode microphysical properties becomes extremely unstable or even impossible. For other parameters, the results degrade but remain useful in the sense that deviations from the true values mostly remain smaller than a priori uncertainties. In future, the situation could be improved by extending the spectral range and by linking parameters of different aerosol size modes (e.g. assuming a common refractive index for both fine and coarse modes).

The ability of RAPSODI to retrieve all atmospheric species simultaneously in a single model atmosphere has some advantages: on the one hand, the algorithm exploits additional information on aerosol contained in the trace gas dSCDs (increasing the DOFS of aerosol profiles by about $20\%$). Conversely, this approach for the first time propagates the uncertainties of the





**Table 7.** Approximate relative changes in information (DOFS) and accuracy (RMSD between retrieval results and true values) between non-polarimetric (Multi-S mode) and polarimetric (Multi-S-P) retrievals from elevation-scan observations. Positive numbers indicate improvements (increase in DOFS or a decrease in RMSD).

|  |  | Idealised[a] [%] | Increased noise[b] [%] |
|---|---|---|---|
| Information (DOFS) | Aerosol profiles | 62 | 28 |
|  | HCHO profiles | 25 | -7 |
|  | NO$_2$ profiles | 19 | -3 |
|  | Aerosol properties | 82 | 66 |
| Accuracy (RMSD) | Aerosol VCD | 70 | 55 |
|  | HCHO VCD | 14 | -39 |
|  | NO$_2$ VCD | 25 | -39 |
|  | Aerosol conc.[c] | 67 | 49 |
|  | HCHO conc.[c] | 71 | 32 |
|  | NO$_2$ conc.[c] | 73 | 56 |
|  | Aerosol properties | 42 | 38 |

[a] Identical measurement noise for non-polarimetric and polarimetric observations
[b] Increased noise for polarimetric observations
[c] Values refer to the concentrations in the surface layer

light path constraining aerosol abundances and aerosol properties into the trace gas profiles. As a consequence, the information on the latter is reduced by about $15\,\%$.

Finally, it should be noted that results presented in this study depend on a priori assumptions. The DOFS is a measure of the information gain relative to the a priori knowledge (see Equation 18). In addition, the a priori biases of the retrieval results are expected to change with the choice of $\mathbf{x}_a$ and $\mathbf{S}_a$. In an ideal case, $\mathbf{S}_a$ is calculated by inferring expected variability as well as retrieval parameter cross-correlations from climatologies. These are obviously space- and time-dependent and the same applies to $\mathbf{S}_a$. Furthermore, $\mathbf{S}_a$ often contains some arbitrary component, which may be introduced as a result of simplifying assumptions, and sometimes $\mathbf{S}_a$ is tweaked to prevent divergence of the inversion. The impact of the choice of $\mathbf{S}_a$ on the obtained DOFS is estimated in Figure S15 in the supplement. Our findings suggest that, even though the numbers may change, the major conclusions drawn in this study remain valid.

# 13 Outlook

Although our studies with synthetic data provide a first analysis for the potential of polarimetric MAX-DOAS measurements, the jury is still out until real observations have been thoroughly investigated and validated. To this end, we have constructed





and operated a PMAX-DOAS instrument as described in Section 3, and we have taken a series of measurements at the Hohenpeißenberg site in southern Germany. Presentation of the campaign results are out of scope for the present paper. However, they are discussed in detail in Tirpitz (2021) and we summarize the major findings here: indeed, the first evaluations were successful and confirm our findings regarding the information content analysis in Section 7. However, validation of the retrieval results turned out to be difficult due to the lack of representative and accurate independent observations. Furthermore, there

are indications that some of the assumptions made in the RAPSODI algorithm are too simple: in particular the assumption of a Lambertian surface albedo and vertically homogeneous aerosol properties might be critical when performing polarimetric retrievals from field data. Another important aspect is the impact of clouds. While MAX-DOAS inversions of trace gas VCDs and surface concentrations from non-polarimetric dSCDs are surprisingly robust under cloudy conditions (e.g. Frieß et al., 2011; Frieß et al., 2016), corresponding investigations for the retrieval of the full state vector (according to Section 5.4) from

polarimetric observations still need to be performed. It is not yet clear to what extent and in what way clouds affect observed polarimetric dSOTs and dSCDs, nor is it known how reliably the individual state vector elements can be retrieved under cloudy conditions. Also different cloud filtering approaches might be investigated to improve such retrievals on the cost of available data.

*Code and data availability.*  The RAPSODI algorithm and synthetic dataset will be made publicly available prior to the final publication of

the paper.

*Author contributions.*  JLT implemented the RAPSODI retrieval algorithm, performed the presented investigations and wrote the first draft of the manuscript. UF initiated the investigations and contributed to all activities in his role as the primary supervisor of the project. RS optimised the VLIDORT software package for the purposes described and provided support with its use. UP contributed to the conceptualisation, scientific discussion and interpretation. All authors contributed to the manuscript.

*Competing interests.*  The authors hereby declare that they have no conflict of interest.

*Acknowledgements.*  JLT acknowledges Prof. Dr. André Butz and Prof. Dr. Klaus Pfeilsticker for fruitful discussions on inversion theory. This work has been performed as part of the German Research Association (DFG) Project RAPSODI (Project No. 272342164). We further acknowledge funding by the Heidelberg Graduate School for Physics and by Prof. Dr. André Butz. We acknowledge financial support by the Open Access Publishing Fund of Ruprecht-Karls-Universität Heidelberg.





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
