# Peer review of "Enhancing MAX-DOAS atmospheric state retrievals by multispectral polarimetry - studies using synthetic data"

_Atmospheric Measurement Techniques, 2021_

## Author Comment (AC1)

**Authors' responses to referee comments**

**Introduction**

We thank the referees for their comments; these have helped a lot to improve the manuscript.

In the following:

- Original referee comments are printed in **bold face**
- Our answers are printed in standard font
- Explicit changes applied to the manuscript are printed in *italic font.*

**Answers to referee #1**

**p. 11, l.309 The approach to model the aerosol bulk extinction efficiency to be constant with altitude limits the possibility to retrieve vertical aerosol profiles with different aerosol types at different altitudes (e.g. a mixture of soot, nitrate aerosols and sea salt in the lower part of the atmosphere and desert dust at higher altitudes). This is a limitation that could be mentioned in the manuscript.**

We agree. It should be noted that RAPSODI (which does not make use of Eq. 21) can perform retrievals with different aerosol types at different altitudes. However, for the present study we have assumed constant aerosol properties. We have added the following sentence in line 291:

*"RAPSODI has the ability to define and retrieve up to three different aerosol types, which can be assigned to different atmospheric layers."*

And again at line 310, we have added

*"Note that Eq. 21 only applies if aerosol properties are assumed to be constant with altitude, as is the case in the present study."*

**Fig. 3. Please add to the legend the meaning of the horizontal lines corresponding to each histogram (e.g. median)**

Adopted, see answers to referee #2 for details.

**p. 23, l.485 ".... if measurements are performed at only two ... " please rewrite "a", e.g. "at" / "for".**

Adopted.

**p. 27, l.546-547 It could be considered to leave out the last part of this sentence because of redundancy: "(the effective ... polarimetric measurements)"**

We have changed the text in brackets from:

*"(the effective loss of light is only a factor of two, and there are three times as many polarimetric measurements)"*

To:

*"(even though the effective loss of light is only by a factor of two)"*

**Answers to referee #2**

**P1, L2: "analysis ultra-violet" ) "analysis of ultra-violet"**

Adopted.

**P1, L7 & L8: "measurement" ) "measurements"**

Adopted.

**P1, L12: "retrieval aerosol" ) "retrieval of aerosol"**

Adopted.

**P8, L234: The aim of inverse model ) inverse modelling**

Adopted.

**Table 1: Since 360nm and 477nm are used to cover the O4 absorption and 343nm and 460nm are more or less central wavelengths for HCHO and NO2 respectively, I was wondering why did you use 415nm and 532nm? What is the benefit and how would the neglection of these wavelengths deteriorate the results?**

The additional wavelengths 415 nm and 532 nm were added to ensure a more-or-less spectrally equidistant sampling of the dSOT, in order to incorporate broadband polarisation and intensity information into the retrieval (see Section 5.2 for further explanation). Six wavelengths turned out to be sufficient to capture this information (compare the orders of typical DOAS broadband fitting polynomials). When using fewer wavelengths, the loss of information becomes significant and increases with each wavelength taken out. If the simulations at 415 nm and 532 nm are omitted, e.g. for the Multi-S-P mode, the total DOFS values decrease by 11%.

To make things clearer, we have inserted the following sentence in the paper:

*"The use of two additional wavelengths (415 nm and 532 nm) ensures an equidistant spectral sampling of dSOTs, in order to incorporate the full broadband polarisation and intensity information into the retrieval."*

**P13, L337: Why did you decide for an exponential grid? Most retrieval algorithms use an equidistant spacing in all retrieval altitudes.**

We have added a sentence in the paper which explains the motivation behind this approach:

*"The exponential grid reflects the fact that the sensitivity and vertical resolving power of ground-based MAX-DOAS observations decrease with altitude. Its advantages are (1) that the information content is more evenly distributed over the retrieved layers, and (2) that accurate RT-simulation results can be obtained with fewer simulation layers (and thus lower computational cost) if an exponential instead of a regular grid is applied (see also Tirpitz, 2021)."*

We note also that many older MAX-DOAS retrieval algorithms use irregular grids (at least for the radiative transfer simulations), in that the atmosphere is subdivided into multiple regular sub-grids, in which layer thicknesses increase with altitude.

**P17, L401: DOFS in table S2 are larger so I would write larger instead of smaller here.**

It should be noted that we are talking about DOFS increases between different scenarios, not about the DOFS values themselves. To make this clearer, we have changed the corresponding line from:

*"… increases in DOFS for the concentration profiles are significantly smaller…"*

To:

*"…for the concentration profiles, changes in DOFS between different measurement modes are significantly smaller…"*

**Table 6: I was wondering if e.g. the UV only mode means that for the trace gases only one wavelength is used but for aerosols both?**

> No. The idea was that the UV mode corresponds to a conventional retrieval, meaning that three separate retrieval runs are performed for each species:
>
> 1. Retrieval of aerosol profile and properties from $O_4$ dSCDs in the dominant absorption band at 360 nm only(!).
> 2. Retrieval of an $NO_2$ profile from $NO_2$ dSCDs at 360 nm
> 3. Retrieval of an HCHO profile from HCHO dSCDs at 343 nm

**If yes, how large is the DOFS for individual parameters of aerosols at different O4 absorption bands?**

As mentioned before, this was not investigated for the O4 absorption band at 343 nm, only for the bands at 360 nm and 477 nm. Results for individual aerosol parameters can be found in Table 6 of the manuscript (look at UV-only and Vis-only modes, if you are especially interested in the conventional retrievals)

**I would assume that the aerosol information content of the absorption band at 343 nm is extremely small and the benefit of including this wavelength is insignificant.**
Yes, we agree, and this is why we did not include this band.

**Fig. 3: What do the thick vertical lines mean?**

> Apologies for the incomplete description. We have added the following to the figure description:
>
> *"Thick vertical lines indicate median values."*

**Please add also rows for Vis.**

Adopted.

**Why do trace gases show double peaks for Multi-S-P-A and Multi-S-P-A-I? Do we see different results for HCHO and $NO_2$ here? Please discuss!**

In the original manuscript, DOFS for $NO_2$ and HCHO have been summed up to obtain the orange histograms. To make things clearer, we have changed the figure to show separate histograms for HCHO and $NO_2$ and to improve visibility, we have multiplied the corresponding DOFS values by a factor of two to displace them with respect to the other histograms. The new figure is:

[Figure]

We have modified the figure caption accordingly. The double peaks in the histograms arise from the different aerosol scenarios chosen for this study. We have added the following paragraph to the discussion:

*"The trace gas histograms in Figure 3 for the Multi-S-P and Multi-S-P-A-I show double peaks, which appear to be related to the choice of aerosol scenarios. Those scenarios with high aerosol loading close to the surface (Aer3, Aer4, and Aer5) have reduced sensitivity to trace gases at lower altitudes and mostly contribute to the left peak of the histogram, whereas other aerosol scenarios mostly accumulate in the right peak."*

**Fig. 4: This is the first time you show box-whisker plots. Please give a short explanation in the caption of this figure.**

We have added the following to the Figure description:

*"In this plot, box-whiskers represent the distribution of observed values: boxes encompass values between the 25th and 75th percentile, while whiskers indicate the 5th and 95th percentiles. Horizontal line markers within the boxes indicate median values."*

**P21, L444-L445: Could you please explain more about where the sensitivity for the surface albedo comes from? You do not look to the surface so photons must be scattered into line of sight mainly due to multiple scattering.**

Exactly. We have added the following explanatory sentence:

*"The albedo sensitivity mostly originates from observations at low EAs, where a significant fraction of detected light experiences reflection from the depolarising Lambertian surface before being scattered*

*into the line of sight. In conventional non-polarimetric aerosol retrievals from $O_4$ dSCDs, this effect is barely noticeable, since the resulting light path enhancement is small. In contrast, the observed polarisation state is strongly affected and thus very sensitive to the surface properties."*

**Would a negative elevation angle increase the sensitivity?**

We assume that the referee refers here to the sensitivity of the polarisation state w.r.t. the surface albedo; the answer to this question is yes. An elevated instrument measuring at shallow negative elevations (resulting in light paths of the order of 1 km between the instrument and the LOS-surface-intersection) will result in the highest sensitivity of the polarisation state to the surface albedo. If light paths between instrument and surface are shorter (larger negative elevation angles), the sensitivity will decrease again, because most of the detected light comes directly from the depolarising surface (with limited contributions of polarised light from atmospheric scattering). However, we believe that discussing this in detail is outside the scope of the paper.

**Fig. 9 and 10: It would be interesting to compare with what the normal uv and vis mode would retrieve since this is what the community is mostly doing. Could you please add this as well or are results similar to UV-S?**

The last point applies: the differences between UV-only (conventional retrieval) and UV-S are very small (see also Table 6) and barely noticeable in Figures such as Fig. 9 and 10. We have focussed on the S-modes here, because the improvement through polarisation is clearly visible and distinguishable from other new features of the RAPSODI algorithm.

**Could you please change the noiseless lines in a different way? It is hard to identify them.**

For the thin curves we chose a dotted line style and increased the transparency. E.g. Fig 9 now looks like this:

[Figure]

**What does the shaded area around the a priori mean? The caption indicates that this is the uncertainty of a priori?**

> Exactly.

> **I thought one fix a priori is used? Please explain!**

Yes, the a priori uncertainty is the same for all modes. It appears to differ because the scales of the axes change.

**Fig. 11**: **I am surprised by the VCD results. In general, common MAX-DOAS profiling algorithms perform strongly for integrated quantities but might have larger deviations for certain profile features. If possible, please add a discussion and maybe another column to Table 6 for VCDs.**

Particularly for the UV-S mode, the VCD results in our study deviate much further from the truth than is the case in former studies. This issue has been discussed already in the original manuscript on P26, L523-526: the reason is that not only do we retrieve vertical profiles but we also retrieve surface and aerosol properties. The problem is thus strongly under-determined and we currently believe that (for the UV-S and Multi-S mode) the retrieval ends up in local minima of the cost function. In fact, if we fix surface and aerosol properties to their true values, we obtain accuracies comparable to those in former studies.

Calculating DOFS for VCDs is not trivial. Among other things, the DOFs are ambiguous, since it is necessary to define how any perturbations of the VCD are distributed into individual layers. Their usefulness is therefore limited; Table 6 already contains a large number of values, and we have therefore decided to not include VCD DOFS in an additional column.

**Section 11: This section is interesting but I would assume even larger uncertainties for real data. Especially when thinking about measurements of elevation scans at different azimuthal directions and subsequent Almucantar scans, temporal changes in the atmospheric composition might vary strongly. Which part of Kreher et al. 2020 supports 5% and why do you assume that this is enough based on the study of synthetic data by Frieß et al. 2019?**

We agree with this comment. Frieß et al. 2019 chose 5%, inspired by the dSCD RMSD between different MAX-DOAS instruments during the CINDI-2 intercomparison campaign. These RMSDs are reported by Kreher et al. 2020 e.g. in Table 7. Considering that the CINDI-2 instruments were measuring synchronously and were aligned to a common viewing direction, we admit that this is not a very robust empirical basis and the 5% variation is likely too optimistic. It should also be noted that the variability will strongly depend on location and conditions. Corresponding extensive investigations in this direction need to be performed in the future.

We have changed the corresponding paragraph from:

*"To put the noise magnitudes into perspective: Frieß, 2019 for instance assume an inhomogeneity noise of 5%, motivated by observations performed during the CINDI-2 campaign (Kreher, 2020)."*

To:

*"To put the noise magnitudes into perspective: Frieß, 2019 for instance assume an inhomogeneity noise of 5% for their synthetic studies, inspired by the dSCD RMSD observed between different MAX-DOAS instruments during the CINDI-2 campaign (Kreher, 2020). However, depending on location and conditions, variabilities can be much larger."*

**Fig. S14: Please change colour and name of "Multi-S-P-A-I" mode in legend. And add the description of random noise magnitudes for the x-axis in the caption.**

Adopted. We have added horizontal grey dotted lines to indicate a priori values (as in Fig. 11)

**Table 7: Please add another column and discussion of the data presented in section 11 for 6% because this analysis is most likely closest to real measurements.**

As mentioned in the manuscript, results for the ancillary study described in section 11 were calculated for a single atmospheric scenario (Aer1-TG1) and therefore cannot be compared to the other results shown in Table 7. Calculations for the full dataset will require significant additional computational effort, which we consider to be disproportionate in this context. Furthermore, as discussed in the previous points, it is not clear which magnitude of variability is closest to reality. We have therefore decided to not include this data in Table 7.

**References**

Tirpitz, J.-L.: Enhancing MAX-DOAS atmospheric remote sensing by multispectral polarimetry, PhD Thesis, University of Heidelberg, https://doi.org/10.11588/heidok.00030159, 2021

Frieß, U., et al.: Intercomparison of MAX-DOAS vertical profile retrieval algorithms: studies using synthetic data, Atmospheric Measurement Techniques, 2019, 12, 2155-2181

Kreher, K. et al.: Intercomparison of $NO_2$, $O_4$, $O_3$ and HCHO slant column measurements by MAX-DOAS and zenith-sky UV-visible spectrometers during CINDI-2, Atmospheric Measurement Techniques, 2020, 13, 2169-2208